# Temperature-dependent fasciation mutants provide a link between mitochondrial RNA processing and lateral root morphogenesis

Kurataka Otsuka[1†‡], Akihito Mamiya[1†], Mineko Konishi[1§], Mamoru Nozaki[1#], Atsuko Kinoshita[1¶], Hiroaki Tamaki[1**], Masaki Arita[1], Masato Saito[1††], Kayoko Yamamoto[1‡‡], Takushi Hachiya[2], Ko Noguchi[3], Takashi Ueda[4], Yusuke Yagi[5], Takehito Kobayashi[5§§], Takahiro Nakamura[5], Yasushi Sato[6], Takashi Hirayama[7], Munetaka Sugiyama[1*]

[1]Botanical Gardens, Graduate School of Science, The University of Tokyo, Tokyo, Japan; [2]Department of Molecular and Functional Genomics, Interdisciplinary Center for Science Research, Shimane University, Shimane, Japan; [3]Department of Applied Life Science, School of Life Sciences, Tokyo University of Pharmacy and Life Sciences, Tokyo, Japan; [4]Division of Cellular Dynamics, National Institute for Basic Biology, Aichi, Japan; [5]Department of Bioscience and Biotechnology, Faculty of Agriculture, Kyushu University, Fukuoka, Japan; [6]Biology and Environmental Science, Graduate School of Science and Engineering, Ehime University, Ehime, Japan; [7]Institute of Plant Science and Resources, Okayama University, Okayama, Japan

**\*For correspondence:**
sugiyama@ns.bg.s.u-tokyo.ac.jp

[†]These authors contributed equally to this work

Present address: [‡] Division of Molecular and Cellular Medicine, National Cancer CenterResearch Institute, Tokyo, Japan / R&D Division,Kewpie Corporation Sengawa Kewport, Tokyo, Japan / Division of Molecular and Cellular Medicine, Institute of MedicalScience, Tokyo Medical University, Tokyo, Japan; [§] Biotechnology Research Center, The University of Tokyo, Tokyo, Japan; [#] Biotechnology Research Center and Department of Biotechnology, ToyamaPrefectural University, Toyama, Japan; [¶] Department of Biological Sciences, Graduate School of Science, TokyoMetropolitan University, Tokyo, Japan; [**] Health and Crop Sciences Research Laboratory, Sumitomo Chemical Co.Ltd, Hyogo, Japan; [††] Innovation Promotion Division, Oji Holdings Corporation, Tokyo, Japan; [‡‡] Department of Biological Sciences, Graduate School of Science, Tokyo, Japan; [§§] GRA&GREEN Inc, Incubation Facility 106, Nagoya University, Aichi, Japan

**Competing interests:** The authors declare that no competing interests exist.

**Abstract** Although mechanisms that activate organogenesis in plants are well established, much less is known about the subsequent fine-tuning of cell proliferation, which is crucial for creating properly structured and sized organs. Here we show, through analysis of temperature-dependent fasciation (TDF) mutants of Arabidopsis, *root redifferentiation defective 1* (*rrd1*), *rrd2*, and *root initiation defective 4* (*rid4*), that mitochondrial RNA processing is required for limiting cell division during early lateral root (LR) organogenesis. These mutants formed abnormally broadened (i.e. fasciated) LRs under high-temperature conditions due to extra cell division. All TDF proteins localized to mitochondria, where they were found to participate in RNA processing: RRD1 in mRNA deadenylation, and RRD2 and RID4 in mRNA editing. Further analysis suggested that LR fasciation in the TDF mutants is triggered by reactive oxygen species generation caused by defective mitochondrial respiration. Our findings provide novel clues for the physiological significance of mitochondrial activities in plant organogenesis.

## Introduction

Plants elaborate their architecture by continuously developing new organs, such as leaves, floral organs, axillary stems, and lateral roots (LRs). Organogenesis begins with the local activation of cell proliferation in the plant body. In the following stages, proliferation is restricted to certain areas, which is essential for the formation of properly sized and structured organs. However, the molecular underpinnings of such regulation remain mostly unknown.

LRs serve as building blocks of the root system architecture and are crucial for the uptake and transport of water and minerals. The first visible step of LR formation occurs within the parent root,

where a few cells start to divide, comprising the LR primordium. The LR primordium grows and eventually emerges out of the parent root to form a new LR (*Torres-Martínez et al., 2019*). This process has been described in detail in the model plant *Arabidopsis thaliana* (Arabidopsis), rendering it one of the most ideal systems to study the molecular mechanisms of organ development (*Goh et al., 2016*; *von Wangenheim et al., 2016*). In Arabidopsis, a small number of cells in a few adjacent files of the xylem pole pericycle layer, termed LR founder cells, first divide in the anticlinal (orthogonal to the proximodistal axis of the primary root) orientation (*Figure 1B*; *Goh et al., 2016*; *von Wangenheim et al., 2016*). The local accumulation of the phytohormone auxin is critical for LR initiation, driving LR founder cell identity acquisition and division via the degradation of the SOLITARY ROOT (SLR/IAA14) repressor, thus activating the expression of downstream genes mediated by the AUXIN RESPONSE FACTORS ARF7 and ARF19 (*Lavenus et al., 2013*). However, much less is understood about the coordinated periclinal (parallel to the proximodistal axis of the root) and anticlinal divisions that subsequently take place. In particular, the manner in which cell proliferation becomes confined to the central zone of the primordium, giving rise to the dome-shaped structure, largely remains a mystery (*Torres-Martínez et al., 2019*), although the requirement of several factors, such as polar auxin transport (*Benková et al., 2003*; *Geldner et al., 2004*), control of auxin response (*De Smet et al., 2010*), peptide hormones (*De Smet et al., 2008*; *Murphy et al., 2016*), transcription factors (*Du and Scheres, 2017*; *Hirota et al., 2007*), symplastic connectivity (*Benitez-Alfonso et al., 2013*), epigenetic gene regulation (*Napsucialy-Mendivil et al., 2014*), and mechanical interaction with the overlaying tissue (*Vermeer et al., 2014*), has been revealed.

*root redifferentiation defective 1* (*rrd1*), *rrd2*, and *root initiation defective 4* (*rid4*) are temperature-sensitive mutants of Arabidopsis that were originally isolated by us via screening using adventitious root (AR) formation from hypocotyl tissue segments as an index phenotype (*Konishi and Sugiyama, 2003*; *Sugiyama, 2003*). In addition to AR formation, other aspects of development, such as seedling growth and callus formation, were affected by high-temperature conditions (*Konishi and Sugiyama, 2003*; *Sugiyama, 2003*). Most notable among these aspects was their LR phenotype, in which abnormally broadened (i.e. fasciated) LRs were formed at 28°C (non-permissive temperature), but not at 22°C (permissive temperature), in a tissue culture setting; thus, we termed the three mutants as temperature-dependent fasciation (TDF) mutants (*Otsuka and Sugiyama, 2012*). It was later revealed that the early stages of LR development are likely affected in the TDF mutants, and that the fasciated LRs exhibit exclusive enlargement of inner tissues (*Otsuka and Sugiyama, 2012*), suggesting that the genes responsible for the TDF mutations (TDF genes) encode negative regulators of cell division that are important for the size restriction of the central zone during the formation of early stage LR primordia; however, their molecular identity has remained elusive.

Plant cells have gene expression systems in mitochondria and plastids in addition to the nucleus. Although organelle gene expression is typically associated with organelle-specific functions, it might also be involved in higher order physiological activities including the regulation of organogenesis. Mitochondria are considered the 'powerhouses' of the cell, as they supply the energy that is necessary for cellular activities. In comparison with other eukaryotes, RNA metabolism in mitochondria is particularly complex in plants, and entails numerous nuclearly encoded RNA-binding proteins (*Hammani and Giegé, 2014*). Given the relaxed nature of transcription, post-transcriptional processing, such as RNA editing, splicing, maturation of transcript ends and RNA degradation, are known to play predominant roles in shaping the plant mitochondrial transcriptome (*Hammani and Giegé, 2014*). Many factors that participate in plant mitochondrial RNA processing have been identified; however, the implications of their role in regulating plant organ development remain unclear (*Hammani and Giegé, 2014*).

Herein, we report a detailed analysis of the TDF mutants. We found that LR fasciation in the TDF mutants was caused by extra cell division in the early stages of LR formation. Next, we identified all three TDF genes as encoding nuclearly encoded mitochondrial RNA processing factors. Analysis of mitochondrial RNA demonstrated that RRD1 is involved in the removal of poly(A) tails, and that both RRD2 and RID4 are RNA editing factors. Defective protein composition of the mitochondrial electron transport chain was found in *rrd2* and *rid4*. Phenocopying of the TDF mutants by mitochondrial respiratory inhibition and reactive oxygen species (ROS) induction, together with its reversal by ROS scavenging, suggested that ROS generation resulting from impaired RNA processing is the primary cause of the extra cell division observed during early LR development in the TDF mutants. Our

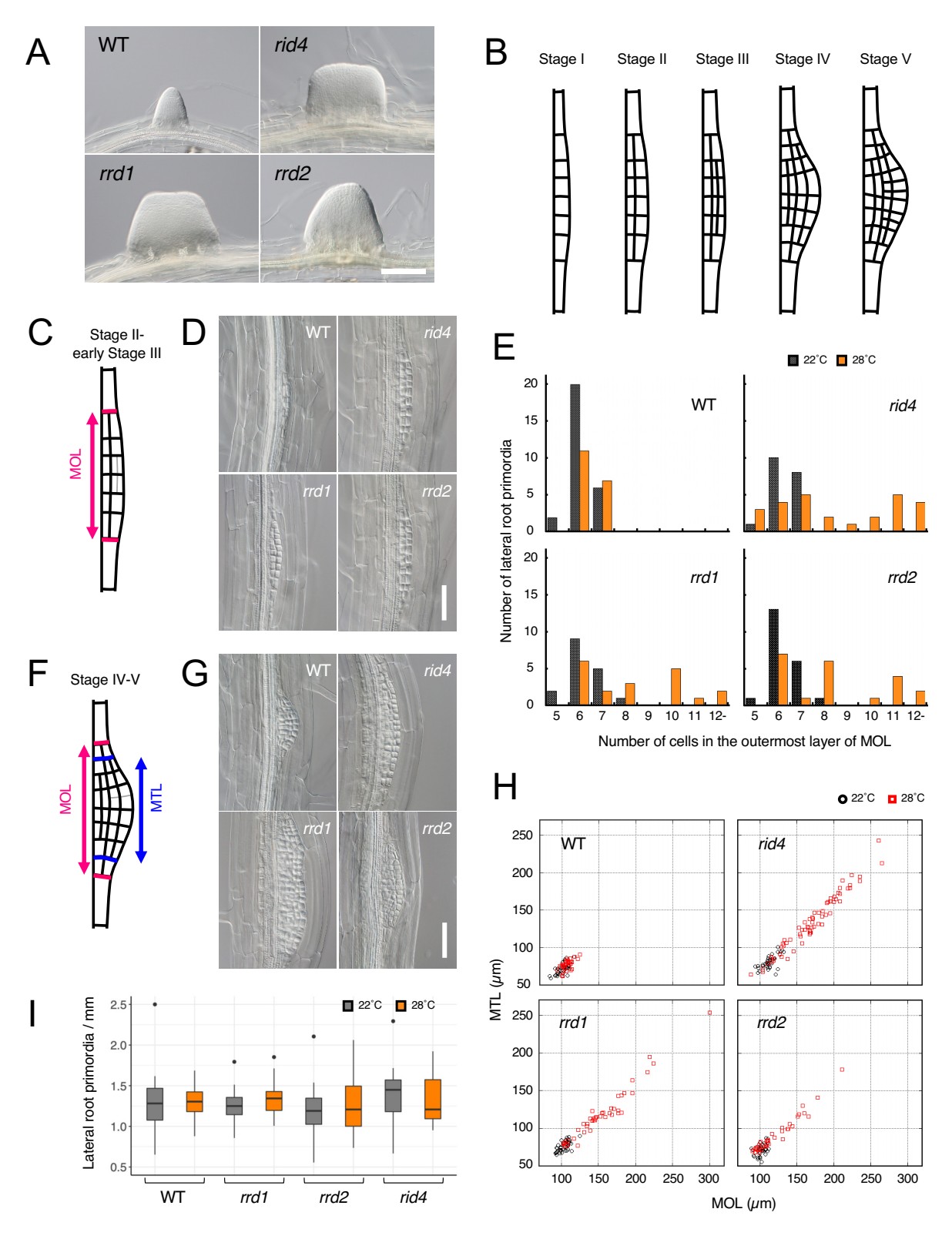

**Figure 1.** Effects of the temperature-dependent fasciation (TDF) mutations on the early stages of lateral root (LR) development. (A) Fasciated LRs formed at 28°C in the TDF mutant explants vs. a normal root on the wild-type (WT) explant after 6 days of culture. (B) Schematic representation of LR development (stages I–V). (C) Schematic image of a primordium at stage II. The area consisting of more than one cell layer (MOL) is delimited by red lines. (D) Stage II primordia formed at 28°C in WT and TDF mutant explants. (E) Effects of the TDF mutations on the number of cells in the outermost

*Figure 1 continued on next page*

*Figure 1 continued*

layer of the MOL area of stage II primordia at 22°C (black) and 28°C (orange). N = 17–28. (F) Schematic image of a primordium at the transition from stage IV to stage V. The areas consisting of MOL and more than two cell layers (MTL) are delimited by red lines and blue lines, respectively. (G) Stage IV–V primordia formed at 28°C in WT explants and TDF mutant explants. (H) Scatterplot of the effect of the TDF mutations on the width of the MTL vs. the width of the MOL areas at 22°C (black) and 28°C (red). N = 31–66. (I) LR densities in the WT explants and TDF mutant explants cultured at 22°C or 28°C (including all developmental stages; median, 25–75% quantile, N = 21–29, p>0.3, Kruskal-Wallis test). Scale bars, 100 μm (A), 50 μm (D, G).

The online version of this article includes the following source data for figure 1:

**Source data 1.** Raw data and supplement to transparent reporting form for *Figure 1I*.

discovery shed light on a new aspect of mitochondrial RNA processing that is relevant in the control of plant organogenesis.

## Results

### Effects of the TDF mutations on LR formation

To gain insight into fasciated LR formation in the TDF mutants, a detailed investigation was carried out using the semi-synchronous LR induction system (*Ohtani et al., 2010*), in which nearly de novo LR formation is induced from root explants of young seedlings upon culture in auxin-containing root inducing medium (RIM). In this system, a 6-day culture of TDF mutant explants results in high rates of LR fasciation at 28°C (non-permissive temperature) (*Figure 1A*), but not at 22°C (permissive temperature) (*Otsuka and Sugiyama, 2012*). To determine the stage of LR formation at which developmental abnormalities occur in the TDF mutants, LR primordia from earlier time points were examined. In Arabidopsis, LR formation begins with anticlinal cell divisions in the xylem pole pericycle cell file, producing an array of short cells flanked by longer cells, which serve as the origin of the LR primordium (stage I; *Figure 1B*; *Goh et al., 2016*; *von Wangenheim et al., 2016*). This is followed by periclinal divisions throughout the primordium, with the exception of the flanking cells in some occasions, creating two cell layers (stage II; *Figure 1B*). Subsequent periclinal cell divisions take place in the central zone of the primordium, producing the third cell layer (stage III), followed by the fourth cell layer (stage IV; *Figure 1B*). Additional anticlinal cell division, together with cell expansion at the innermost cell layer, gives rise to a dome-shaped primordium (stage V; *Figure 1B*). The comparison of the number of cells within the area consisting of more than one layer (MOL) (*Hirota et al., 2007*) between stages II and III, revealed that all TDF mutants showed an increase in this parameter in a temperature-dependent manner (*Figure 1*, C–E). The same trend was observed in primordia at stage IV and V, for which the widths of the MOL and more than three layer (MTL) (*Hirota et al., 2007*) areas were quantified (*Figure 1*, F–H). These results showed that the LRs of TDF mutants contain more basal cells in the initial steps of LR development, namely as early as stage II, than normal LRs and indicated that the increase in the number of cells along the lateral axis of the primordium induces the expansion of its central zone, giving rise to an abnormally broadened and flat-shaped LR. As there was no significant increase in LR density (*Figure 1I*; Kruskal-Wallis test, p>0.3), LR fasciation in the TDF mutants seems to be the result of the expansion of individual primordia, as opposed to the fusion of multiple primordia because of overcrowding that is observed in some other mutants (*Benitez-Alfonso et al., 2013*; *De Smet et al., 2008*).

### Positional cloning and expression analysis of the TDF genes

To clone the TDF genes, we mapped the mutated loci in the TDF mutants based on the temperature-sensitive AR formation phenotype, which originally led to the isolation of the mutants (*Figure 2—figure supplement 1*; *Konishi and Sugiyama, 2003*; *Sugiyama, 2003*). The candidate genes identified by sequencing the mapped regions were confirmed either by a complementation test (*RRD1* and *RID4*; *Figure 2—figure supplement 2, A and E*) or an allelism test (*RRD2*; *Figure 2—figure supplement 2, B* to D). This resulted in the identification of *RRD1* as At3g25430, which encodes a poly(A)-specific ribonuclease (PARN)-like protein, and *RRD2* and *RID4* as At1g32415 and At2g33680, respectively, both of which encode a pentatricopeptide repeat (PPR) protein belonging to the PLS subfamily (*Figure 2A*). At1g32415 has previously been reported as the gene responsible for the *cell wall maintainer 2* (*cwm2*) mutation (*Hu et al., 2016*); thus, we will refer to it as *RRD2/*

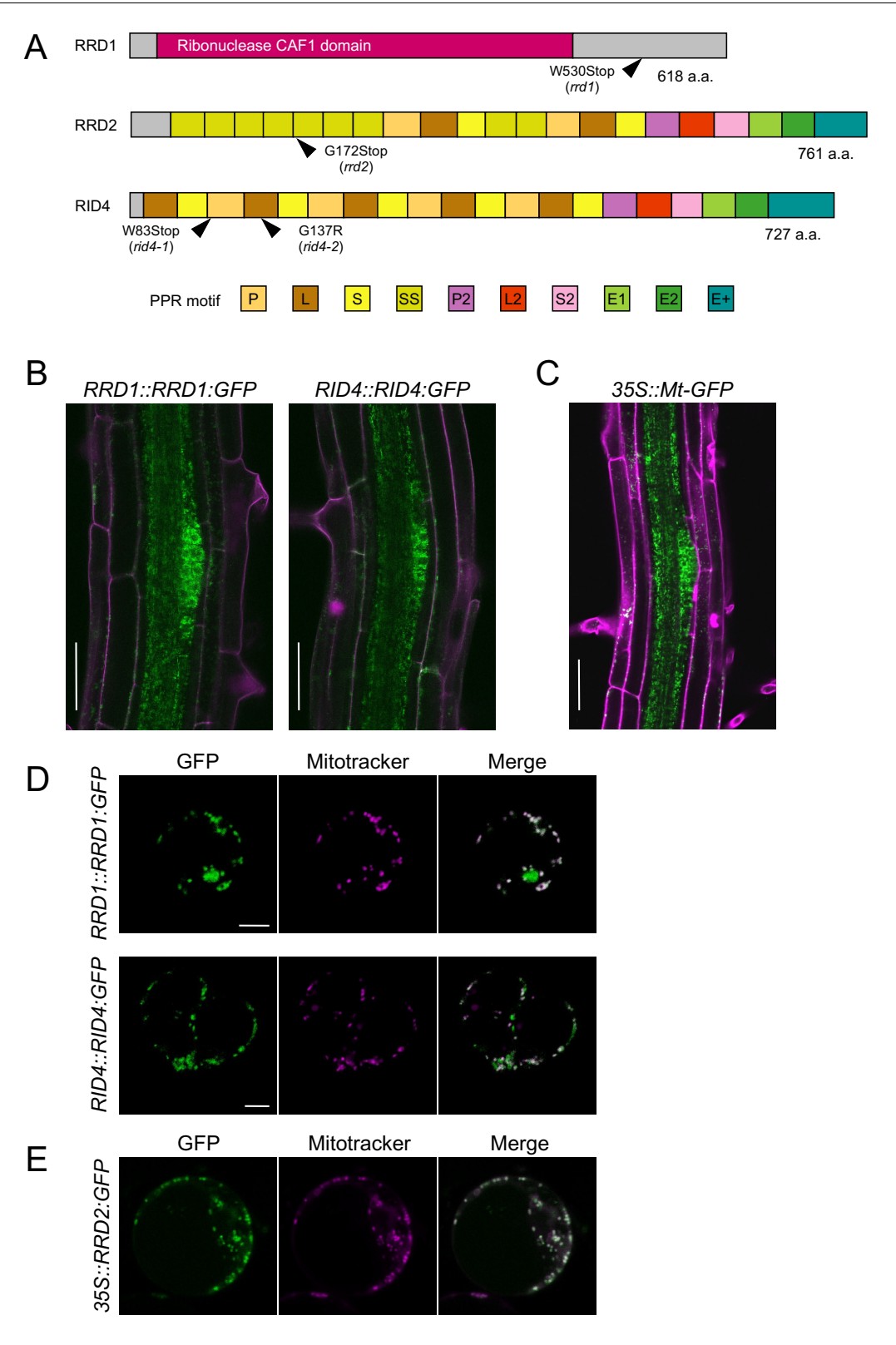

**Figure 2.** Tissue-specific expression and subcellular localization of the temperature-dependent fasciation (TDF) proteins. (A) Structures of the RRD1, RRD2, and RID4 proteins. (B and C) Expression of *RRD1::RRD1:GFP* (B, left), *RID4::RID4:GFP* (B, right), and *35S::Mt-GFP* (C) at stage II of lateral root (LR) primordium development. Propidium iodide was used as a red counterstain. (D and E) Expression of *RRD1::RRD1:GFP* (D, upper panels), *RID4::RID4:*

*Figure 2 continued*

GFP (**D**, lower panels), and *35S::RRD2:GFP* (**E**) in callus-derived protoplasts. Mitochondria were labeled with MitoTracker Orange. Scale bars, 50 μm (**B** and **C**) and 5 μm (**D** and **E**).

The online version of this article includes the following figure supplement(s) for figure 2:

**Figure supplement 1.** Chromosome mapping of the temperature-dependent fasciation (TDF) mutations, *rrd1*, *rrd2*, and *rid4-1*.

**Figure supplement 2.** Complementation analysis and allelism test for the identification of the temperature-dependent fasciation (TDF) genes *RRD1*, *RRD2*, and *RID4*.

**Figure supplement 3.** Identification and characterization of the *rid4-2* mutant.

**Figure supplement 4.** Functionality and expression of *RRD1::RRD1:GFP* and *RID4::RID4:GFP*.

**Figure supplement 5.** Colocalization of chlorophyll autofluorescence and *RRD1::RRD1:GFP* and *RID4::RID4:GFP*.

*CWM2* henceforth. *rrd1*, *rrd2*, and *rid4-1* are all nonsense mutations (*Figure 2A*). The *rrd1* mutation results in an 89-amino-acid C-terminal truncation of the 618-amino-acid RRD1 protein; the mutant protein may be partially or conditionally functional. As the *rrd2* and *rid4* mutations create a stop codon close to the start codon (*Figure 2A*), they are likely to eliminate gene function. Later in our study, another mutant harboring a mutation in the *RID4* gene was isolated and designated *rid4-2* (*Figure 2A* and *Figure 2—figure supplement 3*). *rid4-2* exhibited LR fasciation as well as retarded seedling growth at high-temperature conditions, similar to *rid4-1* (*Figure 2—figure supplement 3, A and B*). The *rid4-2* mutation is a missense mutation that gives rise to a single amino acid substitution (G137R) (*Figure 2A* and *Figure 2—figure supplement 3D*), presumably causing a partial reduction of gene function.

GFP reporter studies were carried out to elucidate the expression patterns of the TDF genes. For *RRD1* and *RID4*, genomic constructs encompassing the promoter region to the end of the protein-coding sequence (*RRD1::RRD1:GFP* and *RID4::RID4:GFP*) were generated and introduced into *rrd1* and *rid4-1*, respectively. The suppression of the mutant AR phenotype demonstrated the functionality of the reporter genes (*Figure 2—figure supplement 4, A and B*). For both *RRD1* and *RID4*, strong GFP expression was mostly confined to apical meristems and LR primordia in the root system and slightly and much weaker expressions were detected in the stele and cortex/epidermis tissues, respectively (*Figure 2B*, and *Figure 2—figure supplement 4C*). This resembled the *35S::Mt-GFP* line, which expresses mitochondria-targeted GFP under the constitutive active cauliflower mosaic virus (CaMV) 35S promoter (*Figure 2C*). At the subcellular level, in callus-derived protoplasts, fluorescence from the GFP-fusion proteins appeared punctate or granulated and was largely overlapped with signals from the mitochondrion-specific dye MitoTracker Orange, demonstrating that the majority of RRD1 and RID4 proteins are localized to mitochondria (*Figure 2D*). We also examined the overlap of GFP signal with chlorophyll autofluorescence in protoplasts prepared from greening-induced callus and found that a small fraction of RRD1 resides in the plastid as well (*Figure 2—figure supplement 5*). Although the tissue-level investigation of *RRD2/CWM2* expression was unsuccessful because of the undetectable levels of the signals of *RRD2::RRD2:GFP*, mitochondrial localization was also confirmed for RRD2 by studying transient expression under the 35S promoter (*Figure 2E*). Together, these data showed that the TDF genes *RRD1*, *RRD2/CWM2*, and *RID4* encode putative RNA processing factors that localize to mitochondria.

## Analysis of the role of RRD1 in poly(A) degradation of mitochondrial mRNAs

PARN belongs to the DEDD superfamily of deadenylases (*Pavlopoulou et al., 2013*). Recent human and animal studies have led to an increased appreciation of its participation in the maturation process of a wide variety of noncoding RNAs (*Lee et al., 2019*). In plants, however, PARN plays a distinct role in the removal of the poly(A) tails of mitochondrial mRNA (*Hirayama et al., 2013*; *Hirayama, 2014*; *Kanazawa et al., 2020*). Given the sequence similarity to PARN and its mitochondrial localization, we hypothesized that RRD1 is also involved in regulating the poly(A) status of mitochondrial mRNA. To test this possibility, we first performed a microarray analysis of poly(A)$^+$ RNAs prepared from wild-type and *rrd1* explants that had been induced to form LRs at 28°C, and found a substantial increase in mitochondria-encoded poly(A)$^+$ transcripts in *rrd1* explants (*Figure 3A*, and *Figure 3—figure supplement 1, A* to C). As the majority of plant mitochondrial transcripts normally lack poly(A) tails, presumably because of swift removal after its addition (*Holec et al., 2008*), we

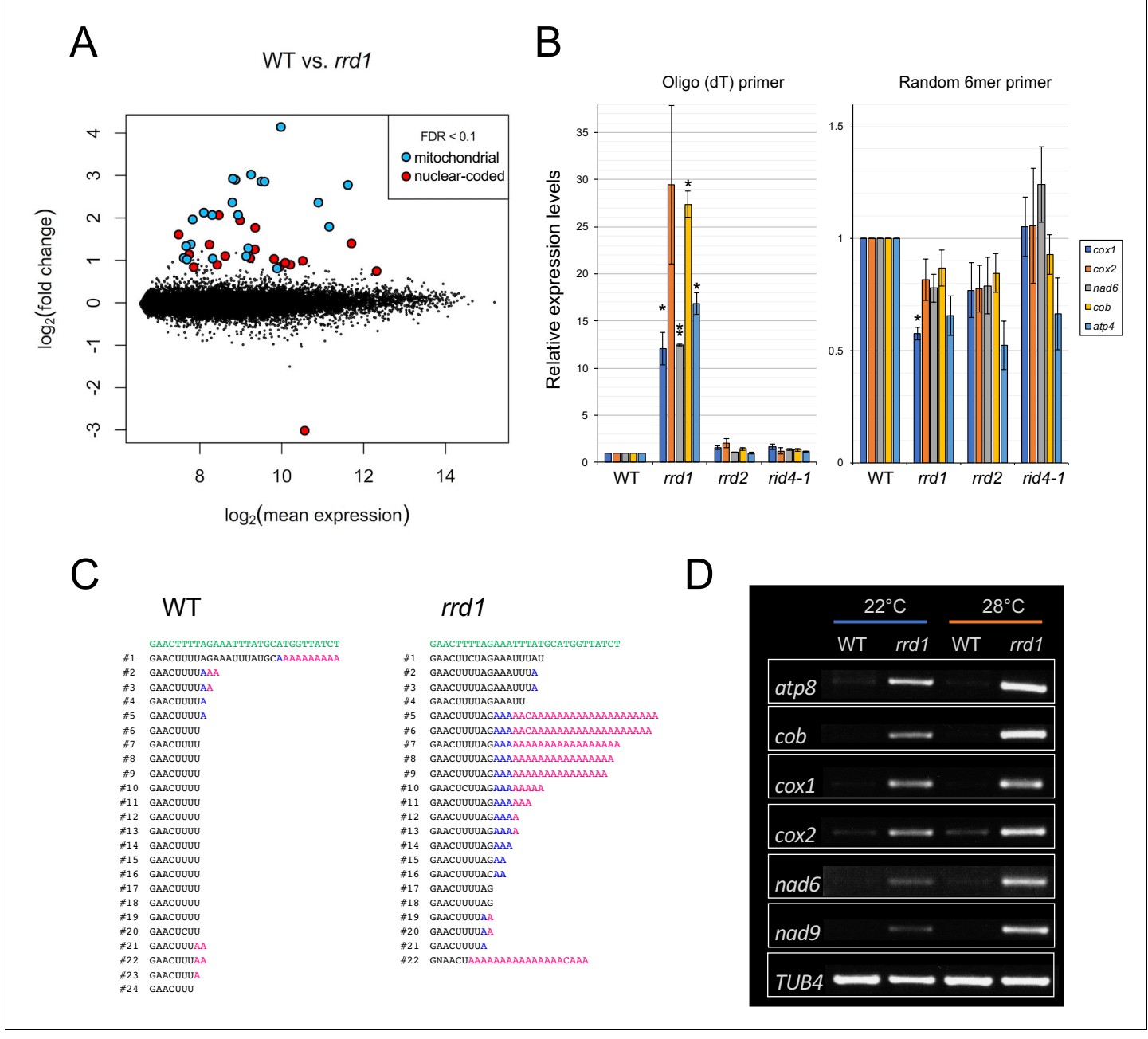

**Figure 3.** Accumulation of polyadenylated mitochondrial transcripts in *rrd1*. (**A**) MA plot for the microarray analysis of poly(A)+ transcripts of *rrd1* vs. wild-type (WT) explants in which lateral roots (LRs) were induced at 28°C for 12 hr. (**B**) qRT–PCR analysis of explants in which LRs were induced at 28°C for 12 hr. The total and polyadenylated transcript levels are shown for *cytochrome oxidase subunit 1* (*cox1*), *cox2*, *NADH dehydrogenase subunit 6* (*nad6*), *apocytochrome B* (*cob*), and *ATP synthase subunit 4* (*atp4*) (mean ± s.d., N = 3, *p<0.05, **p<0.01, one sample *t* test with Benjamini-Hochberg correction). (**C**) Analysis of the 3′ end of the *cox1* mRNA by CR–RT PCR. mRNAs were prepared from WT and *rrd1* seedlings that were first grown at 22°C for 7 days, and then at 28°C for 2 days. The genomic sequence of *cox1* is shown in green. (**D**) RACE-PAT assay showing the accumulation of polyadenylated transcripts of *atp8*, *cob*, *cox1*, *cox2*, *nad6*, *nad9*, and *TUB4*. mRNAs were prepared from explants in which LRs were induced at 22°C or 28°C for 12 hr.

The online version of this article includes the following source data and figure supplement(s) for figure 3:

**Source data 1.** Raw data and supplement to transparent reporting form for *Figure 3B*.
**Figure supplement 1.** Characterization of RRD1 function.
**Figure supplement 2.** Sequence alignment between RRD1 and PARNs from various organisms.

suspected that the apparent sharp increase in mitochondrial transcript level might be ascribed to defective poly(A) tail removal, rather than increased transcription. In fact, a comparative analysis of polyadenylated and total RNA levels via quantitative reverse transcription polymerase chain reaction (qRT-PCR) revealed a selective increase in polyadenylated transcripts (*Figure 3B*). Furthermore, a circularized RNA (CR)-RT PCR analysis (*Forner et al., 2007*) of the *cytochrome oxidase subunit 1* (*cox1*) mRNA was performed to study its 3′ extremity, and revealed a marked increase in the polyadenylated to non-polyadenylated ratio in *rrd1* compared with the wild-type plant (*Figure 3C*). In addition, a poly(A) test assay by rapid amplification of cDNA ends (RACE-PAT) (*Sallés et al., 1999*) showed that polyadenylated transcript levels were increased at higher temperature in *rrd1* (*Figure 3D*). Taken together, these results demonstrated that RRD1 is involved in poly(A) tail removal in mitochondrial mRNAs, and that, in *rrd1*, polyadenylated mitochondrial transcripts accumulate in a temperature-dependent manner.

Next, we investigated whether the RRD1 protein itself has deadenylation activity. In previous studies, this possibility was excluded because, in contrast to canonical PARNs (including AtPARN/ AHG2), RRD1 lacks three out of the four amino acids that are essential for its function as a deadenylase (*Figure 3—figure supplement 2*; *Reverdatto et al., 2004*). In our assay, as expected, the recombinant RRD1 protein did not show any activity in the conditions effective for human PARN (*Figure 3—figure supplement 1, D and E*). We concluded that the RRD1 protein alone does not have deadenylase activity.

To assess the effects of the observed accumulation of poly(A)$^+$ mitochondrial transcripts in *rrd1*, we introduced the *ahg2-1 suppressor 1* (*ags1*) mutation into *rrd1*. *ags1* is a mutation of a mitochondrion-localized poly(A) polymerase (PAP), AGS1, which was originally identified based on its ability to counteract compromised function of AtPARN/AHG2 (*Hirayama et al., 2013*). A substantial decrease in mitochondrial poly(A)$^+$ transcript levels was observed in the *rrd1 ags1* double mutant compared with the *rrd1 AGS1* control (*Figure 4A*). Moreover, *rrd1* phenotypes, such as temperature-dependent LR fasciation and shoot and root growth retardation of seedlings (*Sugiyama, 2003*), were significantly alleviated (*Figure 4*, B and C). These results indicate that the accumulation of poly(A)$^+$ mitochondrial transcripts is the primary cause of the *rrd1* phenotype.

## Analysis of the roles of RRD2 and RID4 in mitochondrial mRNA editing

PPR proteins are known for their role in regulating various aspects of organellar post-transcriptional gene expression, such as RNA stabilization, RNA cleavage, RNA splicing, RNA editing, and translation (*Barkan and Small, 2014*; *Hammani and Giegé, 2014*). They are characterized by the tandem assembly of degenerate protein motifs of about 35 amino acids, termed PPR motifs (*Barkan and Small, 2014*). The PPR motifs allow PPR proteins to recognize specific sites of single-stranded RNAs through a one-motif to one-base interaction (*Barkan and Small, 2014*). The PPR protein family has undergone a remarkable expansion in land plants, representing one of the largest protein families thereof (*Barkan and Small, 2014*). RRD2 and RID4 belong to the PLS-class of PPR proteins, most of which have been reported as being C-to-U RNA editing factors (*Kobayashi et al., 2019*). The PLS class PPR proteins contain three types of PPR motifs, the P motif (normally 35 a. a. in length), the L motif (35–36 a. a. (long)) and the S motif (31 a. a. (short)), in contrast to the P-class PPR proteins, which only contain P motifs (*Barkan and Small, 2014*; *Cheng et al., 2016*). Considering their localization to mitochondria (*Figure 2*, D and E), we speculated on the involvement of RRD2 and RID4 in the editing of mitochondrial RNA. A comprehensive sequence analysis of previously reported RNA editing sites using cDNA prepared from explants induced to form LRs at 28℃ revealed an almost complete abolishment of C-to-U editing at two sites (*cytochrome c biogenesis protein 2* (*ccb2*)−71C and *ccb3*-575C) in *rrd2* and at six sites (*ATP synthase subunit 4* (*atp4*)−395C, *ribosomal protein l5* (*rpl5*)−58C, *rpl5*-59C, *rps3*-1344C, *rps4*-77C, and *rps4*-332C) in *rid4* (*Figure 5A*, *Figure 5—figure supplements 1* and *3*). The identification of *ccb3*-575C as an RRD2/CWM2 editing site was in agreement with a previous study of *cwm2* (*Hu et al., 2016*). Editing was also completely abolished in these sites at 22℃ (*Figure 5—figure supplement 4A*). RID4 editing sites showed incomplete editing in *rid4-2*, implying a partial loss of function in this mutant (*Figure 5—figure supplements 1* and *3*). Significant identity was found among the 5′ upstream sequences of the editing sites that were affected in each mutant (*Figure 5—figure supplement 4B*), further suggesting that RRD2 and RID4 participate in the editing of these sites via direct contact.

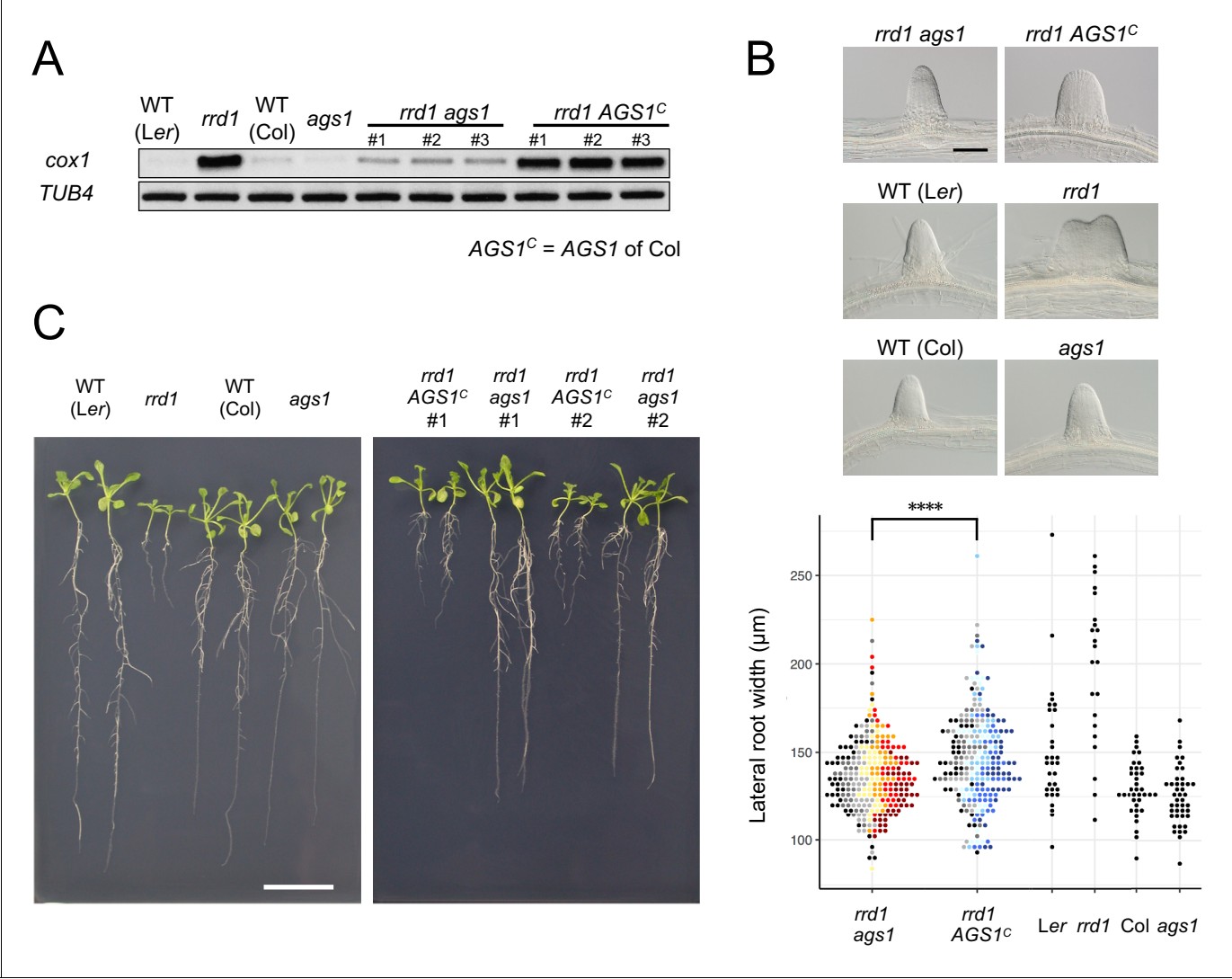

**Figure 4.** Effects of *ags1* on the phenotypes of *rrd1*. (**A**) RACE-PAT assay showing the accumulation of polyadenylated transcripts of *cox1* and *TUB4*. *rrd1* mutant strains harboring either *ags1* or *AGS1ᶜ* (*AGS1* of Col background) were obtained by *rrd1* (L*er* background) × *ags1* (Col background) and *rrd1* × Col crosses, respectively. mRNAs were prepared from seedlings that were first grown at 22°C for 5 days, and then at 28°C for 3 days. (**B**) Representative images of lateral roots (LRs) formed at 28°C after 6 days of culture (upper panels). The basal width of the LRs that were formed in this way was scored (lower panel, N = 115–116 for *rrd1 ags1*, and *rrd1 AGS1ᶜ*, N = 22–43 for others, ****p<10⁻⁴, Mann–Whitney–Wilcoxon test with Bonferroni correction). For *rrd1 ags1* and *rrd1 AGS1ᶜ*, data were gathered from seven strains, which are shown by different colors. (**C**) Seedlings grown at 28°C for 13 days on gellan gum plates. Scale bars, 100 μm (**B**) and 2 cm (**C**).

The online version of this article includes the following source data for figure 4:

**Source data 1.** Supplement to transparent reporting form for *Figure 4B*.

In addition, all editing sites of *ccb3*, with the exception of those that were unedited in the wild type, showed declining levels of RNA editing in both *rrd2* and *rid4* (**Figure 5—figure supplement 1**). However, these sites were not considered as targets of RRD2 and RID4 for the following reasons. These sites were incompletely edited, even in the wild type, as opposed to most other sites (**Figure 5—figure supplement 1**), suggesting that their editing is relatively slow and highly susceptible to fluctuations in the kinetic balance between editing and transcription. Moreover, editing at these sites was almost unaffected at 22°C (**Figure 5—figure supplement 4C**) and was only partially inhibited at 28°C in *rrd2* and *rid4* (**Figure 5—figure supplement 1**), even though these mutants are assumed to have lost the function of the corresponding genes completely. *ccb3*-624C was also not regarded as a target site, despite the complete absence of editing in both *rrd2* and *rid4*, as it was

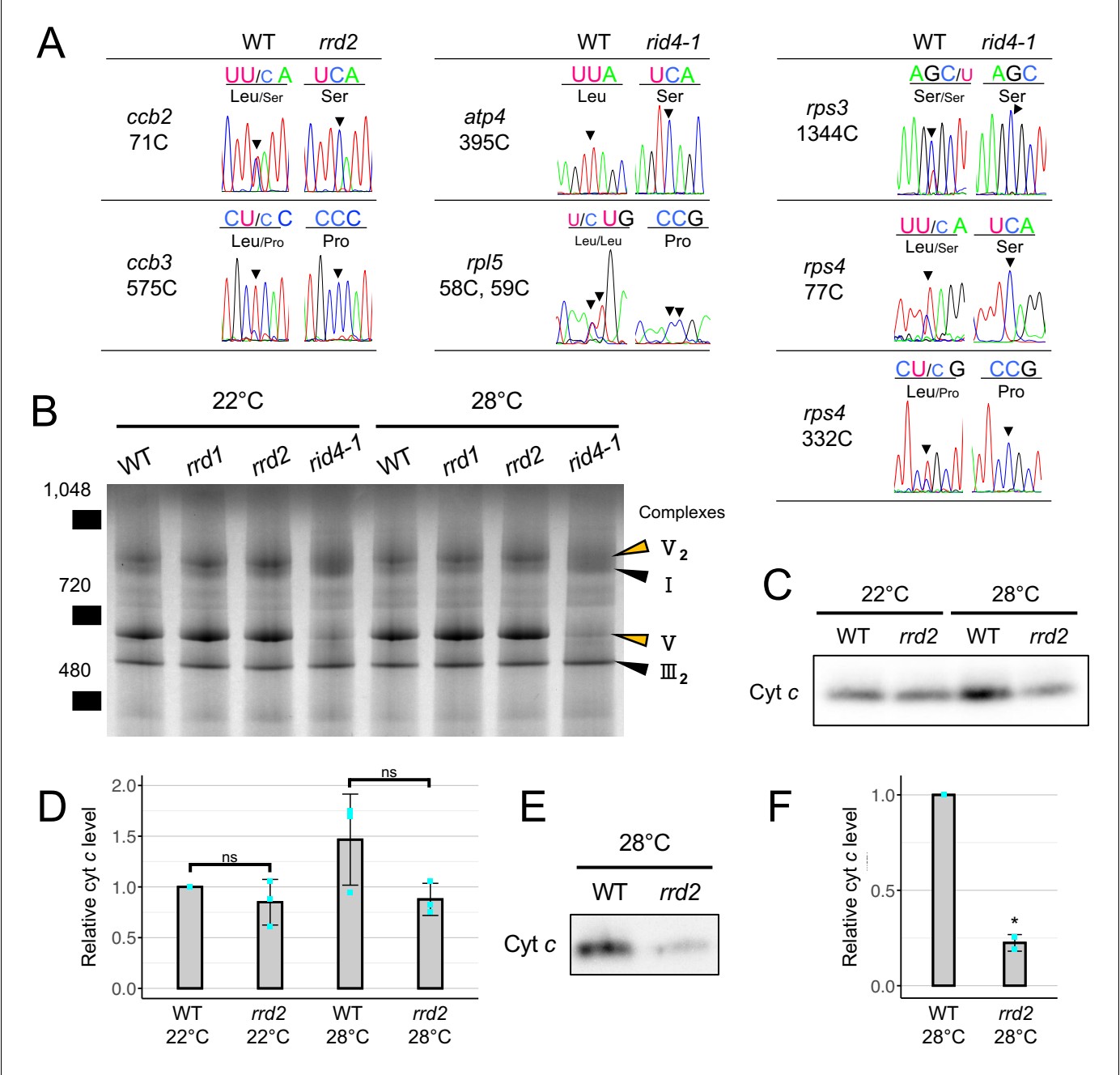

**Figure 5.** Effects of *rrd2* and *rid4* on mitochondrial mRNA editing and protein synthesis. (**A**) Sequencing analysis of mitochondrial mRNA editing in explants in which LRs were induced at 28°C for 12 hr. Arrowheads indicate the RNA editing sites. (**B**) BN-PAGE analysis of mitochondrial protein complexes. Mitochondria were extracted from seed-derived liquid-cultured callus that were first incubated at 22°C for 20 days, and then at 22°C or 28°C for an additional 3 days. Arrowheads indicate the mitochondrial complexes. (**C and D**) Immunoblot analysis of cyt *c*. Mitochondria were extracted in the same conditions as in (**B**). The signal intensities determined by densitometry are shown as values relative to that of the wild type at 22°C in (**D**) (N = 3, mean ± s.d., Welch's *t* test). (**E and F**) Immunoblot analysis of cyt *c* using mitochondria extracted from callus that were cultured first at 22°C for 14 days, and then at 28°C for 7 days. The signal intensities determined by densitometry are shown as values relative to that of the wild type in (**F**) (N = 2, mean ± s.d., **p<0.01, Welch's *t* test).

The online version of this article includes the following source data and figure supplement(s) for figure 5:

**Source data 1.** Raw data and supplement to transparent reporting form for *Figure 5C*.

**Source data 2.** Raw data and supplement to transparent reporting form for *Figure 5E*.

**Figure supplement 1.** Comprehensive analysis of mitochondrial mRNA editing in *rrd2* and *rid4-1* (*atp1* to *cox3*).

**Figure supplement 2.** Comprehensive analysis of mitochondrial mRNA editing in *rrd2* and *rid4-1* (*matR* to *nad7*).

*Figure 5 continued on next page*

*Figure 5 continued*

**Figure supplement 3.** Comprehensive analysis of mitochondrial mRNA editing in *rrd2* and *rid4-1* (*orf240A* to *rps14*(ψ)).
**Figure supplement 4.** Analysis of mitochondrial mRNA editing in *rrd2* and *rid4-1*.

more likely due to originally low levels of editing compared with other sites in *ccb3* (**Figure 5—figure supplement 1**). This view was reinforced by the lack of similarity in the upstream sequence between *ccb3*-624C and the other editing sites that were strongly affected by the *rrd2* and *rid4* mutations (**Figure 5—figure supplement 4B**).

Next, to investigate the effects of losses of function of RRD2/CWM2 and RID4 on mitochondrial protein composition, we performed a blue-native (BN)-PAGE analysis of mitochondrial extracts prepared from seed-derived callus cultured for 3 days at 22℃ or 28℃ after a 20 day 22℃ incubation period. This revealed a substantial loss of complex V (ATP synthase complex) in *rid4* at both 22℃ and 28℃ culture conditions (**Figure 5B**), likely caused by defective mRNA editing of *atp4* (**Figure 5A**), which is a component of this protein complex. No noticeable differences were found in *rrd1* and *rrd2*. Because *ccb2* and *ccb3*, the two mitochondrial genes that are targeted by RRD2/CWM2, are related to cytochrome *c* (cyt *c*) maturation (**Giegé et al., 2008**), we quantified cyt *c* levels in *rrd2*. Cyt *c* levels on a per mitochondrial protein basis were decreased in *rrd2* callus cultured at 28℃ for 3 days (**Figure 5**, C and D) in two out of three cultures, although the difference was not significant when all three results were included. This decrease in cyt *c* levels in *rrd2* was in accordance with a previous analysis of *cwm2* (**Hu et al., 2016**). At 22℃, however, no significant difference was observed between *rrd2* and the wild type. Furthermore, we found that the difference in cyt *c* levels was more pronounced after longer periods of culture at 28℃ (**Figure 5**, E and F). These results indicate that, in *rrd2*, cyt *c* maturation activity was affected to a greater extent at higher temperatures, at least in callus, which possesses root-tissue-like properties, possibly explaining the temperature-dependent nature of its phenotype. The data reported above demonstrated that, in both *rrd2* and *rid4*, the production of certain components of the mitochondrial electron transport chain is hampered by defective mRNA editing.

## Effects of defective mitochondrial respiration on LR formation

Based on the results obtained for *rrd1*, *rrd2*, and *rid4*, we speculated that there might be a relationship between mitochondrial electron transport and cell division control during LR morphogenesis. In fact, the induction of LRs from wild-type explants in the presence of rotenone (complex I inhibitor), antimycin A (complex III inhibitor), or oligomycin (complex V inhibitor) led to LR fasciation, providing evidence that electron transport chain defects are the cause of the TDF LR phenotype (**Figure 6**, A–D). In addition, we found that the effect of antimycin A was stronger at higher temperatures (**Figure 6—figure supplement 1**). To further investigate the underlying molecular pathway, we next asked whether either reduced ATP synthesis, or ROS generation, phenomena that are commonly associated with defective mitochondrial respiration might be involved. We found that the respiratory uncoupler carbonylcyanide m-chlorophenyl-hydrazone (CCCP) did not increase LR width (**Figure 6E**), although LR growth inhibition was observed in a dose-dependent manner (**Figure 6F**), whereas the ROS inducer paraquat (PQ) triggered a significant fasciation of LRs (**Figure 6**, G and H). Furthermore, the application of the ROS scavenger ascorbate resulted in a reversal of the LR broadening induced by PQ treatment (**Figure 6**, G and H). The same effect was observed against the *rid4-2* mutation. These data suggest that the increase in the levels of ROS, but not the decrease in the levels of ATP, acts downstream of defective mitochondrial respiration to promote extra cell division during LR development in the TDF mutants.

Local gradient formation of auxin is important for LR initiation and the subsequent organization of the LR primordium (**Benková et al., 2003**; **Geldner et al., 2004**; **Lavenus et al., 2013**). Strong genetic perturbations of polar auxin transport result in homogeneous proliferation of the pericycle cell layer in large regions of the root upon exogenous auxin treatment. In addition, chemical inhibition of auxin polar transport by naphthylphthalamic acid (NPA) gave rise to broadened LR primordia reminiscent of fasciated LRs of the TDF mutants (**Figure 6—figure supplement 2**). These data indicate a role for local auxin gradient formation in restricting proliferative capacity during LR formation. Therefore, we tested whether ROS-induced LR fasciation is mediated by altered auxin patterning in

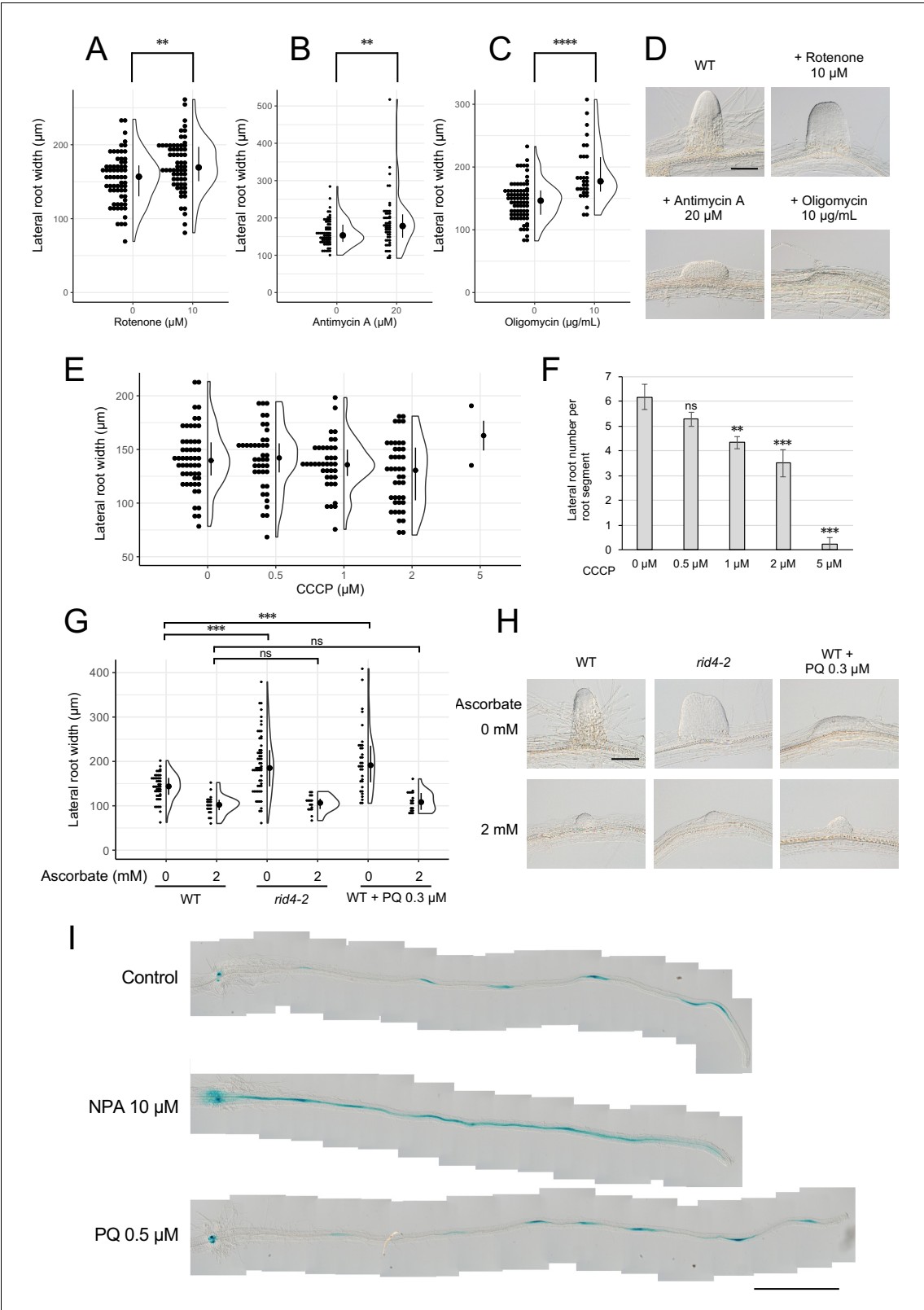

**Figure 6.** Formation of fasciated lateral roots (LRs) after treatment with chemicals that inhibit mitochondrial respiration or induce ROS. (A–D) LRs were induced at 28°C from the wild-type (WT) plant in the presence of rotenone (A), antimycin A (B), or oligomycin (C), and the basal width of the LRs that were formed was scored after 6 days in culture (median, 25–75% quantile, N = 30–76, **p<0.01, ****p<0.0001, Mann–Whitney–Wilcoxon test). Typical LRs that were formed in each treatment are shown in (D). (E and F) LRs were induced from the WT plant in the presence of CCCP. The basal width of

*Figure 6 continued on next page*

Figure 6 continued

LRs (E, median, 25–75% quantile, N = 2–53, p>0.1, Kruskal-Wallis test) and the number of LRs per segment (F, Number of segments = 12, **p<0.01, ***p<0.001, Dunnett's test) were scored on the 6th day. (G and H) The effects of the application of ascorbate on WT, paraquat (PQ)-treated, or rid4-2 segments during LR formation. The basal width of the LRs formed was measured on the 6th day of LR induction (G, median, 25–75% quantile, N = 16–58, ***p<0.001, Mann–Whitney–Wilcoxon test with Bonferroni correction). Representative images of LRs in each condition are shown in (H). (I) DR5::GUS expression at 12 hr after LR induction under treatment with naphthylphthalamic acid (NPA) or PQ. Scale bars, 100 µm (D and H) and 1 mm (I).

The online version of this article includes the following source data and figure supplement(s) for figure 6:

**Source data 1.** Supplement to transparent reporting form for *Figure 6*, A–F.
**Source data 2.** Supplement to transparent reporting form for *Figure 6G*.
**Figure supplement 1.** Effects of antimycin A on lateral root (LR) formation at different temperatures.
**Figure supplement 2.** Effects of naphthylphthalamic acid (NPA) and paraquat (PQ) on lateral root (LR) formation.

early LR primordia. The examination of the expression pattern of the auxin-responsive β-glucuronidase marker *DR5::GUS* (*Benková et al., 2003*; *De Smet et al., 2010*; *Geldner et al., 2004*) at early stages of LR induction, however, did not reveal differences between the control and PQ-treated root segments, whereas treatment with NPA resulted in enhanced expression along the entire root segment (*Figure 6I*). This result indicates that ROS-induced LR fasciation is not caused by an impairment in auxin gradient formation.

## Discussion

In the present study, we investigated three TDF mutants of Arabidopsis, *rrd1*, *rrd2*, and *rid4*, which form fasciated LRs at high temperatures, and identified mutations in previously poorly characterized genes encoding mitochondria-localized proteins as being responsible for the phenotype of these mutants. Our results elucidated the roles of these genes in mitochondrial RNA processing, the construction of the respiratory chain, and in the restrictive control of cell proliferation during LR primordium development.

### Extra cell division during early primordium development leads to LR fasciation

In the present study, we investigated the formation of fasciated LRs observed at high-temperature conditions in the TDF mutants using the semi-synchronous LR induction system (*Ohtani et al., 2010*). By measuring the cell number and primordium width, we found that fasciation of LRs is caused by the excess number of basal cells, which can be detected as early as stage II of LR development (*Figure 1*). The lack of increase in LR density (*Figure 1I*) suggested that LR fasciation is caused by the expansion of individual primordia, rather than the fusion of multiple primordia, which is the case in some other mutants that form abnormally broadened LRs (*Benitez-Alfonso et al., 2013*; *De Smet et al., 2008*). The data are in agreement with the previous result of the temperature-shift experiment, which demonstrated that the first 48 hr following LR induction are critical for LR fasciation in the TDF mutants (*Otsuka and Sugiyama, 2012*), as stage II to early stage III primordia are formed within this time frame (*Figure 2D*; *Ohtani et al., 2010*). The previous characterization of the TDF mutants also showed that fasciated LR primordia exhibit specific enlargement of inner root tissues marked by the expression of *SHORT ROOT* (*SHR*), while the number of cell layers outside the SHR-expressing layer is normal (*Otsuka and Sugiyama, 2012*). A recent study revealed that the area of SHR expression is first established during stage II, where it is confined to the inner layer of the two-cell layered primordium (*Goh et al., 2016*). In subsequent stages, SHR is expressed in cell files derived from the inner layer, which develop into the stele of the LR (*Goh et al., 2016*). Taken together, these results suggest that differentiation into two cell layers at stage II occurs normally in the TDF mutants, and that the increase in the number of cells observed at stage II consequently leads to the expansion of the area of SHR expression in the inner cell layer during LR fasciation.

There are two possibilities that can account for the excess basal cells in LR primordia of TDF mutants: one in which LR founder cells undergo extra rounds of cell division and the other in which extra pericycle cells adjacent to the founder cells are activated to divide. In either case, TDF mutations loosen the restrictive control of cell division and induce some kind of extra cell division. Closer

inspection of the initial process of LR primordium formation by live imaging would distinguish these possibilities.

## RRD1 functions in poly(A) tail removal in mitochondrial mRNA

PARN is a 3′ exoribonuclease of the DEDD superfamily (*Pavlopoulou et al., 2013*), which shows a strong preference for adenine (*Lee et al., 2019*; *Pavlopoulou et al., 2013*). In plants, PARN is involved in the removal of poly(A) tails from mitochondrial transcripts (*Hirayama et al., 2013*; *Hirayama, 2014*; *Kanazawa et al., 2020*). Here, we identified *RRD1* as a gene encoding a PARN-like protein (*Figure 2A*) that resides in mitochondria (*Figure 2, B and C*). Further analysis of *rrd1* demonstrated the participation of RRD1 in poly(A) tail degradation of mitochondrial mRNA (*Figure 3*). In plant mitochondria, immature 3′ extremities of mRNA, together with irregular RNAs, such as 3′ misprocessed mRNAs, rRNA maturation by-products, and cryptic transcripts, are known to be polyadenylated before they are degraded by mitochondrial polynucleotide phosphorylase (mtPNPase) (*Holec et al., 2008*). In fact, down-regulation of mtPNPase in Arabidopsis results in the accumulation of long preprocessed mRNAs, as well as irregular RNAs, the majority of which are polyadenylated (*Holec et al., 2008*). In *rrd1*, unusually long preprocessed mRNAs do not seem to accumulate, as the size of RACE-PAT assay products (*Figure 3D*) corresponded to that of previously reported mature transcript 3′ ends. Total mitochondrial mRNA levels were unelevated in *rrd1* (*Figure 3B*), suggesting that RRD1 is not involved in controlling mRNA abundance by promoting their degradation. Whether by-product accumulation takes place in *rrd1* is not clear. However, given its absence in *ahg2* (*Hirayama et al., 2013*), this is unlikely. Based on these considerations, we concluded that RRD1 plays a distinct role from mtPNPase and seems to be specifically involved in 3′ processing of near-matured mRNA.

The mode of action of the RRD1 protein remains to be solved. The absence of three out of the four catalytic amino acids (DEDD) that are essential for ribonuclease activity (*Figure 3—figure supplement 2*; *Reverdatto et al., 2004*), together with the apparent lack of deadenylase activity of the recombinant RRD1 protein (*Figure 3—figure supplement 1, D and E*), indicated that RRD1 requires additional factors for its participation in poly(A) tail removal.

Failure in the removal of poly(A) tails from mitochondrial transcripts seems to be the primary cause of the *rrd1* phenotype. This is evidenced by the alleviation of the *rrd1* phenotype by the introduction of a mutation of the mitochondria-localized poly(A) polymerase gene *AGS1* (*Figure 4*). As most protein-coding genes in the Arabidopsis mitochondrial genome are involved in the biogenesis of the electron transport chain (*Hammani and Giegé, 2014*), it is likely that mitochondria of *rrd1* carry defects in respiratory activity. However, the exact impact of the altered poly(A) status of mRNAs in mitochondria on electron transport in *rrd1* remains unclear. Unlike the AtPARN/AHG2 loss-of-function mutant *ahg2*, which shows a reduction in complex III levels (*Hirayama et al., 2013*), no significant difference in respiratory chain composition has been detected in *rrd1* to date (*Figure 5B*).

## RRD2 and RID4 function in mitochondrial mRNA editing

Our study identified *RRD2* and *RID4* as At1g32415 and At2g33680, respectively, both of which encode a mitochondria-localized PLS-class PPR protein (*Figure 2*). At1g32415 had previously been reported as the gene responsible for the *cwm2* mutant (*Hu et al., 2016*). A predominant role for PLS-class PPR proteins in RNA editing has been demonstrated with more than 50 out of a total of approximately 200 of these proteins in Arabidopsis having been identified as C-to-U editing factors of mitochondria or plastid RNA (*Kobayashi et al., 2019*). A comprehensive analysis of mitochondrial RNA editing revealed the abolishment of editing at specific sites in *rrd2* and *rid4* (*Figure 5A* and *Figure 5—figure supplements 1*, *2* and *3*). We concluded that both RRD2/CWM2 and RID4 are PLS-class PPR proteins that are involved in mitochondrial mRNA editing.

In *rrd2*, editing at 71C of *ccb2* and 575C of *ccb3* was absent (*Figure 5A*). Both *ccb2* (also known as *ccb206*, *ccmB*, *ABCI2*, and *AtMg00110*) and *ccb3* (also known as *ccb256*, *ccmC*, *ABCI3*, and *AtMg00900*) encode a multisubunit ATP-binding cassette (ABC) protein, which are involved in the maturation of mono hemic *c*-type cytochromes, the soluble cyt *c*, and the membrane-bound cyt $c_1$ of complex III (*Giegé et al., 2008*). Of the two editing sites, *ccb3*-575C was previously reported as a target of *RRD2/CWM2* (*Hu et al., 2016*), whereas *ccb2*-71C is a newly discovered target. A decrease

in the level of cyt *c* was detected in *rrd2*, which is consistent with that reported previously for *cwm2* (*Hu et al., 2016*). The data demonstrated the role of *RRD2/CWM2* in cyt *c* maturation via the RNA editing of cyt *c* biogenesis factors.

In *rid4*, we observed striking reductions in RNA editing at *atp4*-395C, *rpl5*-58, *rpl5*-59C, *rps3*-1344C, *rps4*-77C, and *rps4*-332C (*Figure 5A*). *atp4* (also known as *orf25*, AtMg00640) encodes the peripheral stalk protein (subunit b) of the mitochondrial ATP synthase complex (complex V) (*Heazlewood et al., 2003*). *rpl5*, *rps3*, and *rps4* encode mitochondrial ribosome proteins. Analysis of mitochondrial protein complexes showed a dramatic decrease in the level of complex V in *rid4*, probably because of impaired editing of *atp4*-395C (*Figure 5B*). This is similar to the *organelle transcript processing 87* (*otp87*) mutant of Arabidopsis, in which editing of *atp1*-1178C is deficient (*Takenaka et al., 2019*). These data showed that the formation of complex V could be disrupted by defective RNA editing at a single site of a subunit gene. Considering that the C-to-U editing of the *rps4* transcript at a different site (*rps4*-377) has been shown to affect mitochondrial ribosome assembly in the *growing slowly 1* (*grs1*) mutant (*Takenaka et al., 2019*), it is possible that the *rid4* mutation also has an impact on the mitochondrial ribosome.

Recent advances in the mechanistic understanding of RNA binding by PLS-class PPR proteins have led to the identification of residues at certain positions within the PPR motifs that are important for ribonucleotide recognition (*Barkan and Small, 2014*; *Kobayashi et al., 2019*). By mapping these residues of previously reported RNA-editing PPR proteins to their binding sites, which are located 5' upstream of the editing sites, the so-called 'PPR code' has been elucidated, thus enabling the matching of PPR proteins to their candidate editing targets, and vice versa (*Kobayashi et al., 2019*). According to the recently refined PPR code prediction (*Kobayashi et al., 2019*), RID4 was highly ranked as a potential binding protein of *atp4*-395C (18th, p=4.35 $\times$ $10^{-2}$), *rpl5*-58C (5th, p=3.04 $\times$ $10^{-2}$) and *rps4*-332C (2nd, p=4.06 $\times$ $10^{-3}$). Conversely, these sites were among the predicted editing sites of RID4 (p<0.05) (*Kobayashi et al., 2019*). With regard to RRD2, however, the newly identified binding site (*ccb2*-71C) ranked very low, despite the incorporation of RRD2/CWM2 binding to *ccb3*-575C as learning data for the PPR code prediction (*Kobayashi et al., 2019*). This discrepancy may be related to the unusual arrangement of PPR motifs in RRD2, in which repeats of SS motifs are prevalent, in contrast to canonical PLS-class PPRs, which follow the (P1-L1-S1)$_n$-P2-L2-S2 pattern, such as RID4 (*Figure 2A*; *Cheng et al., 2016*). Nevertheless, given the similarity between the upstream sequences of editing sites which are severely affected by *rrd2* and *rid4* (*Figure 5—figure supplement 4B*), they are likely edited by RRD2 and RID4 via direct interaction. The presented data will contribute to the improvement of PPR protein target estimation.

## The origins of the temperature sensitivity may differ among the TDF mutants

A distinct feature of the TDF phenotype is its exclusive observation at high-temperature conditions (*Konishi and Sugiyama, 2003*; *Otsuka and Sugiyama, 2012*; *Sugiyama, 2003*). Our study revealed some differences in the origin of temperature sensitivity among the TDF mutants. The *rrd1* mutation causes a truncation of the C-terminal domain of the RRD1 protein (*Figure 2A*). This finding, together with the enhancement of poly(A)$^+$ mitochondrial mRNA accumulation at elevated temperatures (*Figure 3D*), implies that, in *rrd1*, RRD1 is partially functional at least at the permissive temperature, and that its activity is more severely affected at the non-permissive temperature. In contrast, the *rrd2* and *rid4-1* mutations introduce a stop codon close to the N-terminus of RRD2 and RID4, respectively, likely resulting in the total loss of their functions (*Figure 2A*). The complete abolishment of RNA editing of the RRD2 and RID4 target sites in the *rrd2* and *rid4-1* mutants, regardless of temperature (*Figure 5A* and *Figure 5—figure supplement 4A*), further supported this idea. However, in *rrd2*, deficient cyt *c* biogenesis was observed only at high temperature (*Figure 5*, C and D). This might be accounted for by the temperature sensitivity of the function of either *ccb2* or *ccb3*, which exhibit alteration of the amino acid sequence in *rrd2*, because of impaired RNA editing (*Figure 5A*).

In *rid4-1*, a huge reduction in complex V biosynthesis was observed both at permissive and non-permissive temperatures (*Figure 5B*). Thus, unlike *rrd1* and *rrd2*, *rid4-1* is constitutively impaired in mitochondrial function. Our study also found that fasciation of LRs observed in the wild type treated with the respiratory inhibitor antimycin A was more pronounced at higher temperatures (*Figure 6—figure supplement 1*). Taken together, these data suggest that mitochondrial impairment and elevated temperature both contribute to LR fasciation in the TDF mutants.

## Impaired mitochondrial electron transport causes LR fasciation likely via ROS production

The phenocopy of the LR fasciation phenotype of the TDF mutants by treatment with respiratory inhibitors demonstrated the causal relationship between defective mitochondrial electron transport and extra cell division during early LR development (*Figure 6*, A–D). Mitochondrial electron transport is best known for its role in driving ATP synthesis through oxidative phosphorylation. Given the lack of LR fasciation after treatment with the mitochondrial uncoupler CCCP (*Figure 6*, E and F), reduced ATP production seems unlikely to be the cause of LR fasciation. The fact that the huge reduction in complex V levels observed in *rid4* (*Figure 5B*) does not lead to LR fasciation at the permissive temperature (*Otsuka and Sugiyama, 2012*) is also supportive of this idea. Experiments using the ROS inducer PQ and the antioxidant ascorbate (*Figure 6*, G and H) pointed to mitochondrial ROS generation as the potential trigger of LR fasciation. A previous study also observed enhanced cell division after the application of another ROS inducer, alloxan, during auxin-induced LR formation (*Pasternak et al., 2005*). In agreement with this 'ROS hypothesis', all three respiratory inhibitors used in our study (rotenone, antimycin A, and oligomycin) are potent inducers of oxidative stress (*Willems et al., 2016*).

ROS have been implicated in stress-induced morphogenic responses (SIMR) (*Potters et al., 2009*). Several studies have shown the involvement of phytohormonal regulation in ROS-triggered SIMR. Altered auxin levels and/or distribution have been proposed as potential mediators in the modulation of cell proliferation in response to oxidative stress (*Pasternak et al., 2005*; *Potters et al., 2009*). Several recent studies have found antagonistic interactions between auxin signaling and mitochondrial ROS (*Huang et al., 2016*). Auxin is a critical factor in LR development, and the centripetal auxin-gradient formation in early-stage LR primordia is thought to contribute to the organization of the LR primordium (*Benková et al., 2003*; *Geldner et al., 2004*). However, neither the pattern nor the intensity of the auxin response visualized by the *DR5::GUS* reporter seemed to be altered under PQ treatment, in contrast to the diffuse pattern observed after the application the auxin polar transport inhibitor NPA (*Figure 6I*). This indicates that ROS-induced LR fasciation is not attributable to a failure in auxin-gradient formation. Further studies of LR fasciation caused by oxidative stress will elucidate novel aspects of the control of cell proliferation during plant organogenesis.

Apart from its role in stress response, ROS has recently emerged as a potential signal in and of itself that is required for plant physiology and development under normal conditions (*Mittler, 2017*). In the primary root, a regulatory mechanism of meristem size involving spatial zoning of different types of apoplastic ROS has been proposed, whereby $O_2^{\bullet-}$ promotes cell proliferation in the meristematic zone, while $H_2O_2$ induces cell differentiation in the elongation zone (*Tsukagoshi et al., 2010*; *Zhou et al., 2020*). The transcription factor UPBEAT1 (UPB1) is suggested to regulate the transition between the two zones, via the suppression of extracellular peroxidase activity in the elongation zone (*Tsukagoshi et al., 2010*; *Zhou et al., 2020*). Interestingly, during LR formation, factors involved in apoplastic ROS regulation, UPB1 and another transcription factor MYB36, as well as some members of the RESPIRATORY BURST OXIDASE HOMOLOG (RBOH) family, are expressed in the periphery of the LR primordium (*Fernández-Marcos et al., 2017*; *Manzano et al., 2014*; *Orman-Ligeza et al., 2016*); however, the role of apoplastic ROS in controlling the proliferation-to-differentiation transition in the LR boundary remains largely speculative. Whether mitochondrial disorders caused by the TDF mutations have an impact on apoplastic ROS also remains to be investigated. In addition, reactive carbonyl species (RCS), which are lipid peroxidation products generated by ROS, were found to mediate auxin signaling in a feed-forward manner during LR formation (*Biswas et al., 2019*; *Mano et al., 2019*); however, no apparent morphological LR phenotype has been observed in RCS-treated plants. The possible involvement of RCS in the TDF LR phenotype awaits further testing.

## Mitochondrial RNA processing is linked to the control of cell proliferation

Mutants of nuclearly encoded mitochondrial RNA processing factors have proven to be useful in probing the physiological roles of mitochondrial gene expression. In particular, studies of C-to-U editing PPR protein genes have led to a collection of about 100 mutants, among which RNA-editing mutants are available for most mitochondrial genes (*Takenaka et al., 2019*). The majority of the

mutations confer visible phenotypes, such as growth retardation, impaired embryo development, late flowering, or reduced pollen sterility (*Takenaka et al., 2019*). Similar developmental defects are also observed in mutants of genes encoding other mitochondrial proteins, including *ndufs4* (complex I mutant), *rpoTmp* (RNA polymerase mutant), and *atphb3* (prohibitin mutant) (*Van Aken et al., 2010*). These results suggest that mitochondria play a supportive role in plant growth, presumably by supplying energy through oxidative phosphorylation. In this study, however, we found that mitochondrial RNA processing is required for preventing extra cell division during LR primordium formation. This suggests that mitochondrial gene expression not only supports active cell proliferation for growth and development but also participates in the local fine-tuning of organ morphogenesis by restricting cell proliferation.

In summary, our study identified an unexpected link between mitochondrial RNA processing and the primordial size control at the early stage of LR development, probably mediated by changes in the level of mitochondrial ROS. This finding provides a novel clue for the physiological significance of mitochondrial activities in the restrictive regulation of cell division required for the proper morphogenesis of plant organs.

## Materials and methods

### Plant materials and growth condition

*Arabidopsis thaliana* (L.) Heynh. ecotypes Columbia (Col) and Landsberg *erecta* (L*er*) were used as Arabidopsis in this work. The TDF mutants *rrd1*, *rrd2*, and *rid4-1* were described previously (*Konishi and Sugiyama, 2003*; *Otsuka and Sugiyama, 2012*; *Sugiyama, 2003*). The *ags1* mutant (*ags1-1*) was also described previously (*Hirayama et al., 2013*). The *35S::Mt-GFP* line was a gift from Shin-ichi Arimura (*Arimura and Tsutsumi, 2002*). *rid4-2* was derived from an ethyl methanesulfonate-mutagenized population of the L*er* strain of Arabidopsis. SALK_027874 was obtained from the Arabidopsis Biological Resource Center. *rrd1* mutant strains harboring either *ags1* or *AGS1^c* were obtained by *rrd1* (L*er* background) × *ags1* (Col background) and *rrd1* × Col crosses, respectively. The *DR5::GUS* line (*Ulmasov et al., 1997*) was a gift from Tom J. Guilfoyle and was crossed three times to L*er* before use. Primers for the genotyping the mutants are listed in *Supplementary file 1*.

For tissue culture experiments, donor plants were aseptically grown on Murashige–Skoog medium supplemented with 1.0% (w/v) sucrose, buffered to pH 5.7 with 0.05% (w/v) 2-morpholinoethanesulfonic acid (MES), and solidified with 1.5% (w/v) agar under continuous light (10–15 μmol m$^{-2}$ s$^{-1}$) at 22°C. For observation of seedling phenotypes, plants were aseptically grown on the same medium solidified with 1.5% (w/v) agar or 0.8% (w/v) gellan gum under continuous light (50–80 μmol m$^{-2}$ s$^{-1}$) at 22°C or 28°C. For self-propagation and crossing, plants were grown on vermiculite under continuous light (approximately 50 μmol m$^{-2}$ s$^{-1}$) at 22°C unless otherwise indicated.

### LR and AR induction

For the induction of semi-synchronous formation of LRs, both the shoot and root tips were removed from 4-day-old seedlings that were grown on agar plates, and the remaining parts were cultured on root-inducing medium (RIM) under continuous light (15–25 μmol m$^{-2}$ s$^{-1}$), as described previously (*Ohtani et al., 2010*). RIM consisted of B5 medium supplemented with 2.0% (w/v) glucose and 0.5 mg l$^{-1}$ indole-3-butyric acid, buffered to pH 5.7 with 0.05% (w/v) MES, and solidified with 0.25% (w/v) gellan gum. Culture temperature was set to 22°C for the permissive condition and to 28°C for the non-permissive condition. For AR induction, hypocotyl segments excised from seedlings were cultured on RIM, as described previously (*Konishi and Sugiyama, 2003*).

### Histological analysis

For whole-mount observation, tissue samples were fixed in 25 mM sodium phosphate buffer (pH 7.0) containing 2% (w/v) formaldehyde and 1% (w/v) glutaraldehyde, rinsed with 100 mM sodium phosphate buffer (pH 7.0), and cleared with an 8:1:2 (w/v/v) mixture of chloral hydrate, glycerin, and water. Observations were made with a microscope equipped with Nomarski optics (BX50-DIC; Olympus) to obtain differential interference contrast (DIC) images.

For morphometric analysis of LR primordia, in order to highlight cell organization, the method of *Malamy and Benfey, 1997* was instead employed for tissue fixation and clearing. Developmental

stages of LR primordia were determined according to *Malamy and Benfey, 1997*. LR primordia at Stages II to early III and at Stages IV to V were chosen from samples that had been collected after 16–24 hr and 24–48 hr of culture in the semi-synchronous root induction system, respectively, and were measured for their width and cell number.

For histochemical detection of GUS reporter expression, tissue samples were fixed in 90% (v/v) acetone overnight at −20°C, rinsed with 100 mM sodium phosphate (pH 7.0), and incubated in X-Gluc solution (0.5 mg ml$^{-1}$ 5-bromo-4-chloro-3-indolyl β-D-glucuronide cyclohexylammonium salt, 0.5 mM potassium ferricyanide, 0.5 mM potassium ferrocyanide, 100 mM sodium phosphate [pH 7.4]) for 140 min at 37°C. After rinsing with 100 mM sodium phosphate buffer (pH 7.0), the samples were mounted on glass slides with an 8:1:2 (w/v/v) mixture of chloral hydrate, glycerin, and water, and then subjected to DIC microscopy.

## Chromosome mapping

The TDF mutants in the L*er* background were crossed with the wild-type Col strain, and the resultant F$_1$ plants were self-pollinated to produce F$_2$ seeds or test-crossed with the mutant plants to produce TC$_1$ seeds. The TC$_2$ lines were then developed by separately collecting self-pollinated progenies from each individual TC$_1$ plant. F$_2$ plant or TC$_2$ lines were checked for the ability of AR formation at 28°C and for DNA polymorphism between L*er* and Col. Chromosome locations of the TDF mutations were determined on the basis of linkage between the mutations and the L*er* alleles of polymorphic marker loci.

## Identification of the TDF genes

Sequencing of the genomic regions to which the TDF mutations were mapped led to identification of candidates of *RRD1*, *RRD2*, and *RID4* as At3g25430, At1g32415, and At2g33680, respectively. Identification of these genes was confirmed by the complementation test or the allelism test as described below.

For the complementation test, genomic clones GL07, encompassing At3g25430 (2.9-kbp 5´-flanking sequence, 2.6-kbp coding sequence, and 2.5-kbp 3´-flanking sequence), and GL91321, encompassing At2g33680 (1.8-kbp 5´-flanking sequence, 3.5-kbp coding sequence, and 2.0-kbp 3´-flanking sequence), were isolated from a transformation-competent genome library (*Ohtani et al., 2010*), and introduced into the *rrd1* and *rid4* mutants, respectively. The resultant transformants were examined for the ability of AR formation at 28°C. To determine allelism between *rrd2* and SALK_027874, which carries a T-DNA insertion in At1g32415, F$_1$ progeny derived by crossing *rrd2* with SALK_027874 was examined for the ability of AR formation at 28°C.

## Plasmid construction

Genomic DNA from L*er* was used as a template for PCR-based amplification of DNA fragments of interest. *RRD1::RRD1:GFP* was constructed by inserting the –2780/+2495 region of the *RRD1* gene (+1 = the first base of the translation initiation codon), which encompassed the genomic region from the promoter to the end of the protein-coding sequence, and the coding sequence of sGFP into pGreen0029 (John Innes Centre). *RID4::RID4:GFP* was similarly constructed by inserting the –2297/+2181 region of the *RID4* gene and the sGFP-coding sequence into pGreen0029. For the construction of *35S::RRD2:GFP*, the +1/+2283 region of the *RRD2* gene was inserted into the pSHO1 vector, a derivative of pHTS13 (*Ueda et al., 2001*). Plasmids for the PARN activity assay were constructed by inserting the coding sequence of RRD1 or human PARN (hPARN) into the pHAT vector (Clontech). The hPARN sequence was derived from the GNP Human cDNA clone IRAK071M01 (RIKEN BioResource Research Center). In this plasmid construction, the N-terminal mitochondrial localization signal (24 a.a.) sequence was deleted from the RRD1 coding sequence, and the SEP-tag C9D sequence (*Kato et al., 2007*) was added to the C-terminus of both RRD1 and hPARN sequences to improve the solubility of these protein products.

## Plant transformation

DNAs such as reporter gene constructs and genomic fragments were transformed into *Agrobacterium tumefaciens* and then into Arabidopsis by the floral dip method (*Clough and Bent, 1998*) or its modified version (*Martinez-Trujillo et al., 2004*). Transgenic plants were selected by antibiotic

resistance and genotyped by PCR for the introduction of the correct transgene. Transient expression of *35S::RRD2:GFP* in protoplasts of cultured cells were done as described in *Ueda et al., 2001*.

## Expression and localization analysis of GFP reporters

Expression patterns of *RRD1* and *RID4* were examined with transgenic plants harboring *RRD1:: RRD1:GFP* and *RID4::RID4:GFP*, respectively. Roots of 6-day-old seedlings of these plants were counterstained with 10 mg l$^{-1}$ of propidium iodide and fluorescence images were obtained using a confocal microscope (FV3000; Olympus). Expression analysis of *35S::Mt-GFP* was performed in the same conditions using a different confocal microscope (FV1200; Olympus). To investigate subcellular localization of the RRD1 and RID4 proteins, protoplasts were prepared from calli that had been induced from the *RRD1::RRD1:GFP* and *RID4::RID4:GFP* explants. The protoplasts were incubated with 100 nM Mitotracker Orange (Invitrogen) for 15 min to visualize mitochondria and then observed using the LSM710 system (Carl Zeiss).

## Microarray analysis and data processing

For microarray analysis, total RNA was extracted with TRIzol reagent (Invitrogen) from explants that had been cultured on RIM for 12 hr in the semi-synchronous LR induction system and purified using the RNeasy microkit (QIAGEN). Affymetrix ATH1 microarrays were hybridized with biotinylated complementary RNA targets prepared from the RNA samples according to the manufacturer's instructions. It should be noted here that all the targets were derived from poly(A)$^{+}$ RNA in principal because the T7-oligo(dT)$_{24}$ primer was used for reverse-transcription at the first step of target preparation. Experiments were performed in biological triplicates. The data sets obtained were processed with a variant of MAS5.0 utilizing robust radius-minimax estimators (*Kohl and Deigner, 2010*). Differential gene expression was identified by RankProd 2.0 (*Del Carratore et al., 2017*). The details of the microarray data was deposited in the Gene Expression Omnibus (http://www.ncbi.nlm.nih.gov/geo/) under accession number GSE34595.

## Analysis of mRNA polyadenylation status with RACE-PAT

RACE-PAT was performed principally according to *Sallés et al., 1999*. Total RNA was extracted with TRIzol reagent (Invitrogen) either from LR-induced explants or seedlings. Total RNA was treated with RNase-free DNase I (Promega) to eliminate genomic DNA, and reverse-transcribed with T7-oligo(dT)$_{24}$ as a primer using the PrimeScript II 1 st strand cDNA Synthesis kit (TaKaRa). Then the poly(A) tail status was analyzed by PCR with a combination of gene-specific and T7 promoter primers. The thermal cycling program consisted of initial 2 min denaturation at 95°C followed by 30 cycles of 20 s at 95°C, 20 s at 57°C, and 10 s at 72°C. Primers for the RACE-PAT are listed in *Supplementary file 1*.

## qRT-PCR analysis

For qRT-PCR, total RNA was extracted with TRIzol reagent (Invitrogen) from explants LR-induced at 28°C for 12 hr. To eliminate genomic DNA, total RNA was treated with RNase-free DNase I (Promega), and reverse-transcribed with a random hexamer or oligo(dT)$_{24}$ primer using SYBR Premix ExTaq II (TaKaRa). qRT-PCR reactions were performed with gene-specific forward and reverse primers using the PrimeScript RT-PCR kit (TaKaRa) on the StepOne Real-Time PCR system (Applied Biosystems). The thermal cycling program consisted of initial 30 s denaturation at 95°C followed by 40 cycles of 5 s at 95°C and 30 s at 60°C. At the end of run, melting curves were established for each PCR product to check the specificity of amplification. Expression levels of mRNAs of interest were normalized relative to *TUBULIN4* (At5g44340) expression. DNA fragments amplified from poly(A)$^{+}$ transcripts of several genes including *cob* were sequenced to check the occurrence of mitochondrial editing, which confirmed that they are derived from the mitochondrial genome but not from their copies present in chromosome 2 (*Stupar et al., 2001*). Experiments were performed in biological triplicates. Primers for the qRT-PCR analysis are listed in *Supplementary file 1*.

## PARN activity assay of recombinant RRD1

The pHAT plasmids in which the RRD1 or hPARN sequence had been inserted were transformed into the Rosetta-gami two strain or the M15 strain of *E. coli*. Colonies were grown overnight at 37°C

in LB medium containing 100 µg ml$^{-1}$ ampicillin and 25 µg ml$^{-1}$ chloramphenicol for Rosetta-gami 2 and 100 µg ml$^{-1}$ ampicillin and 25 µg ml$^{-1}$ kanamycin for M15. The cultures were diluted (6:100) in the same medium and grown at 37°C for approximately 3 hr to reach OD$_{600}$ of 0.3 to 0.4, and then treated with 0.2 mM isopropyl β-D-1-thiogalactopyranoside (IPTG) overnight at 18°C to induce the production of the his-tagged RRD1 and hPARN proteins. After cell lysis, the proteins were purified by TALON Metal Affinity Resin (Clontech) and filtered with Amicon Ultra 0.5 ml (30K; Merck Millipore). For the ribonuclease activity assay, the purified proteins (0.125 mg) or RNase If (1.25 U; NEB) were incubated at 25°C for 60 min with a fluorescent-labeled RNA substrate (5´-fluorescein isothiocyanate (FITC)-CUUUUAG(A$_{20}$); this sequence was derived from the 3´ extremity of *cox1* mRNA (*Figure 3C*)) in 10 µL of reaction medium (1.5 mM MgCl$_2$, 100 mM KCl, 0.1 U RNasin Ribonuclease Inhibitor (Promega), 20 mM HEPES-KOH (pH 7.0), 0.2 mM EDTA, 0.25 mM dithiothreitol, 10% (v/v) glycerol, 0.1% BSA) (*Cheng et al., 2006*). The reaction was stopped by adding an equal volume of gel loading mix (90% formamide, 0.5% (w/v) EDTA, 0.025% (w/v) bromophenol blue) and heating to 90°C for 3 min before cooling on ice. The reaction mixtures were loaded onto a 7 M urea-16% polyacrylamide gel and separated by electrophoresis.

## CR-RT PCR analysis of the 3′ end of mRNA

CR-RT PCR analysis was performed principally according to *Forner et al., 2007*. Total RNA was extracted with TRIzol reagent (Invitrogen) from seedlings that had been cultured for 7 days at 22°C and then 2 days at 28°C. To eliminate genomic DNA, total RNA was treated with DNase I (RT grade; Nippon Gene). Next 1 µg of total RNA was circularized with T4 RNA ligase (Promega), desalted with Amicon Ultra 0.5 ml (10K; Merck Millipore), and then reverse-transcribed with a *cox1* specific primer (Atcox1-1; *Supplementary file 1*) using M-MLV (Moloney Murine Leukemia Virus) Reverse Transcriptase (RNase H minus, point mutant; Promega). The RNA template was degraded by adding 1/5 vol of 1 M NaOH to the reaction mixture and incubating at room temperature for 10 min. The solution was neutralized by adding 1 M HCl and the cDNA was purified with the illustra GFX PCR DNA and Gel Band Purification Kit (GE Healthcare). The 5′−3′ junction sequence was amplified by PCR with *cox1* specific primers Atcox1-5′(−176··−196) and Atcox1-3′(+17··+38) using Ex Taq Hot Start Version (Takara). The thermal cycling program consisted of initial 4 min-denaturation at 95°C, followed by 40 cycles of 20 s at 95°C, 20 s at 50°C, and 40 s at 72°C. The PCR products were purified with the Wizard SV Gel and PCR Clean-Up System (Promega) and cloned into the pGEM-T Easy Vector (Promega) using DNA Ligation Kit <Mighty Mix> (Takara). The constructed vector was transformed into the DH5α strain of *E. coli*, and about 20 clones were sequenced. Primers for the CR RT-PCR analysis are listed in *Supplementary file 1*.

## Analysis of mitochondrial mRNA editing

For the analysis of mitochondrial mRNA editing, total RNA was extracted with TRIzol reagent (Invitrogen) from explants LR-induced at 28°C for 12 hr. Total RNA was treated with RNase-free DNase I (Promega), and reverse-transcribed with a random hexamer using the PrimeScript II 1 st strand cDNA Synthesis kit (TaKaRa). Gene specific primers were used to amplify cDNA by PCR using Ex Taq Hot Start Version (Takara). The thermal cycling program consisted of initial 4 min denaturation at 95°C followed by 30 to 40 cycles of 30 s at 95°C, 30 s at 55°C, and 90 to 120 s at 72°C.

The PCR products were purified either by ExoStar DNA purification reagent (GE Healthcare) or Wizard SV Gel and PCR Clean-Up System (Promega), and then sequenced.

## Analysis of mitochondrial protein

Isolation of intact mitochondria was performed principally according to *Murcha and Whelan, 2015*. Seed-derived callus cultured in liquid callus-inducing medium (CIM) (*Konishi and Sugiyama, 2003*; *Sugiyama, 2003*) in the dark with gentle shaking was used as starting material. About 16 g of callus was homogenized in 40 ml ice-cold grinding buffer (0.3 M Mannitol, 50 mM Tetrasodium pyrophosphate, 2 mM EDTA (Disodium salt), 0.5% (w/v) PVP-40, 0.5% (w/v) BSA, 20 mM L-cysteine, pH 8.0 (HCl)) with a mortar, pestle, and glass beads (0.4 mm diameter). The homogenate was filtered through four layers of Miracloth (Millipore) and centrifuged at 2300 g for 5 min twice. The resulting supernatant was centrifuged at 18,000 g for 10 min. The resulting pellet was resuspended in wash buffer (0.3 M Mannitol, 10 mM *N*-Tris(hydroxymethyl)methyl-2-aminoethanesulfonic acid (TES), 0.1%

(w/v) BSA, pH 7.5 (NaOH)) and layered over a three-step Percoll (GE Healthcare) gradient (40%, 21%, and 16% (v/v)). The gradient was centrifuged at 23,500 rpm (approximately 40,000 g to 70,000 g) for 30 min. Mitochondria were collected from the 21% and 40% interface and washed twice in wash buffer (without BSA) by centrifugation at 18,000 g for 10 min.

For BN-PAGE analysis, 10 µg protein of mitochondria was solubilized in 12 µL Native PAGE Sample Buffer (1% n-dodecyl-β-D-maltoside (DDM), Thermo Fisher Scientific), mixed with 1.8 µL of sample additive (33.3% (w/v) glycerol, 1.67% (w/v) Coomassie Brilliant Blue (CBB) G250), and then separated by electrophoresis on a NativePAGE 4% to 16%, Bis-Tris Gel (Thermo Fisher Scientific). Mitochondrial complexes were identified according to *Eubel et al., 2003*.

For immunoblot analysis, proteins separated via SDS–PAGE were transferred to a PVDF membrane and exposed to a primary antibody against cyt *c* (AS08 343A, Agrisera; 1:5000 dilution). The protein concentrations of the SDS-PAGE samples were adjusted using the XL-Bradford kit (integrale). As a secondary antibody, we used a peroxidase-labeled anti-rabbit antibody (NIF824, GE Healthcare; 1:5000 dilution). Immunodetection was performed by incubating the membranes in the Western BLoT Quant HRP Substrate (Takara) and recording the chemiluminescence by LuminoGraph I (ATTO).

## Graph drawing

Bar charts were drawn using KaleidaGraph version 3.6 (Synergy Software), Excel for Mac (Microsoft), or the ggplot2 package (*Wickham, 2016*) of R software (*R Development Core Team, 2020*). Scatter plots were drawn using KaleidaGraph or the default packages of R software. Dot plots were drawn using the ggplot2 package of R software. Violin plots were overlayed to the dot plots using the geom_flat_violin function developed by Joachim Goedhart (https://gist.github.com/JoachimGoedhart/98ec16c041aab8954083097796c2fe81). Box plots were drawn using the ggplot2 package of R software.

## Acknowledgements

We thank Tsuyoshi Nakagawa for providing the binary vector pGW3, Mamoru Sugita for valuable discussion on the PPR proteins, Hajime Sakurai for the technical support for the expression of *35S:: RRD2:GFP* in protoplasts, Shin-ichi Arimura for providing the *35S::Mt-GFP* line, Yuta Otsuka and Yuki Kondo for the assistance on GFP imaging, Yukiko Sugisawa for the technical support for microarray data collection, Hatsune Morinaka for the assistance on qRT-PCR, and Tom J Guilfoyle for providing the *DR5::GUS* line, and the RIKEN BioResource Research Center for providing the hPARN cDNA clone.

## Additional information

### Funding

| Funder | Grant reference number | Author |
|---|---|---|
| Japan Society for the Promotion of Science | Grants-in-Aid for JSPS Fellows (No. 09J08676) | Kurataka Otsuka |
| Japan Society for the Promotion of Science London | Grants-in-Aid for JSPS Fellows (No. 17J05722) | Akihito Mamiya |
| Ministry of Education, Culture, Sports, Science and Technology | Graduate Program for Leaders in Life Innovation (GPLLI) | Akihito Mamiya |
| Ministry of Education, Culture, Sports, Science and Technology | Grant-in-Aid for Scientific Research on Priority Areas (No. 19060001) | Munetaka Sugiyama |
| Japan Society for the Promotion of Science | Grant-in-Aid for Scientific Research (B) (No. 25291057) | Munetaka Sugiyama |

The funders had no role in study design, data collection and interpretation, or the decision to submit the work for publication.

## Author contributions

Kurataka Otsuka, Conceptualization, Data curation, Funding acquisition, Investigation, Visualization, Methodology, Writing - review and editing, KO designed and performed experiments and data analysis mostly in the first half of this study, including histological analysis of fasciated LRs, positional cloning of *RRD1* and *RRD2*, construction of the reporter genes, subcellular localization analysis of the TDF proteins, microarray data collection, initial analysis of the poly(A) status of mitochondrial mRNAs, and initial pharmacological analysis with respiratory inhibitors; Akihito Mamiya, Conceptualization, Data curation, Software, Formal analysis, Funding acquisition, Validation, Investigation, Visualization, Methodology, Writing - original draft, Writing - review and editing, AM designed and performed experiments and data analysis mostly in the latter half of this study, including expression analysis of the TDF genes, microarray data mining, analysis of polyadenylation and editing of mitochondrial mRNAs, genetic analysis with *ags1*, analysis of mitochondrial proteins, and pharmacological analysis with respiration- and ROS-related drugs; Mineko Konishi, Investigation, Methodology, Writing - review and editing, MK identified *RID4* by positional cloning; Mamoru Nozaki, Investigation, Visualization, Methodology, Writing - review and editing, MN designed and performed analysis of PARN activity of recombinant RRD1; Atsuko Kinoshita, Investigation, Writing - review and editing, AK conducted chromosome mapping of *rrd1* and some of the initial characterization of the TDF phenotype; Hiroaki Tamaki, Investigation, Writing - review and editing, HT isolated the *rid4-2* mutant; Masaki Arita, Investigation, Writing - review and editing, MA conducted chromosome mapping of *rid4-2*; Masato Saito, Investigation, Writing - review and editing, MSa conducted chromosome mapping and genome sequencing of *rid4-2*; Kayoko Yamamoto, Investigation, Writing - review and editing, KY collected preliminary data on the genetic relationship between *rrd1* and *ags1* and performed preliminary analysis of RNA editing; Takushi Hachiya, Investigation, Writing - review and editing, THa contributed to the research design and data interpretation for mitochondrial respiration-related analysis; Ko Noguchi, Formal analysis, Investigation, Writing - review and editing, KN contributed to the research design and data interpretation for mitochondrial respiration-related analysis; Takashi Ueda, Investigation, Methodology, Writing - review and editing, TU contributed to the research design, imaging analysis of GFP reporters, and data interpretation for subcellular localization; Yusuke Yagi, Investigation, Methodology, Writing - review and editing, YY contributed to the research design and data interpretation for RNA editing-related analysis and performed preliminary analysis of RNA editing; Takehito Kobayashi, Investigation, Writing - review and editing, TK contributed to the research design and data interpretation for RNA editing-related analysis; Takahiro Nakamura, Investigation, Methodology, Writing - review and editing, TN contributed to the research design and data interpretation for RNA editing-related analysis and performed preliminary analysis of RNA editing; Yasushi Sato, Investigation, Writing - review and editing, YS contributed to the analysis of the recombinant RRD1 protein; Takashi Hirayama, Conceptualization, Investigation, Writing - review and editing, THi contributed to the research design and data interpretation for RNA metabolism-related analysis; Munetaka Sugiyama, Conceptualization, Resources, Supervision, Funding acquisition, Investigation, Writing - original draft, Project administration, Writing - review and editing, MSu launched the study and conducted preliminary analyses

## Author ORCIDs

Akihito Mamiya (iD) https://orcid.org/0000-0001-6492-7903
Atsuko Kinoshita (iD) https://orcid.org/0000-0001-9095-389X
Ko Noguchi (iD) https://orcid.org/0000-0003-3588-3643
Takashi Ueda (iD) https://orcid.org/0000-0002-5190-892X
Munetaka Sugiyama (iD) https://orcid.org/0000-0002-7050-8964

## Decision letter and Author response

Decision letter https://doi.org/10.7554/eLife.61611.sa1
Author response https://doi.org/10.7554/eLife.61611.sa2

## Additional files

### Supplementary files

- Supplementary file 1. Primers used in this study.

- Transparent reporting form

### Data availability

The microarray data has been deposited in the Gene Expression Omnibus ([http://www.ncbi.nlm.nih.gov/geo/](http://www.ncbi.nlm.nih.gov/geo/)) under accession number GSE34595. All data needed to evaluate the conclusions in the paper are present in the paper and/or the Supplementary Materials.

The following dataset was generated:

| Author(s) | Year | Dataset title | Dataset URL | Database and Identifier |
|---|---|---|---|---|
| Otsuka K, Sugiyama M | 2011 | Transcript profile changes associated with lateral root fasciation in temperature-sensitive mutants | https://www.ncbi.nlm.nih.gov/geo/query/acc.cgi?acc=GSE34595 | NCBI Gene Expression Omnibus, GSE34595 |

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
