## [Decision Letter]

**Acceptance summary:**

This is an excellent manuscript that points to a new and very interesting link between primary metabolism and cell proliferation in lateral roots.

**Decision letter after peer review:**

Thank you for submitting your article "Temperature-dependent fasciation mutants connect mitochondrial RNA processing to control of lateral root morphogenesis" for consideration by *eLife*. Your article has been reviewed by three peer reviewers, and the evaluation has been overseen by Jürgen Kleine-Vehn as the Reviewing Editor and Christian Hardtke as the Senior Editor. The following individuals involved in review of your submission have agreed to reveal their identity: Joseph G Dubrovsky (Reviewer #1); Steffen Vanneste (Reviewer #2); Alexis Maizel (Reviewer #3).

The reviewers have discussed the reviews with one another and the Reviewing Editor has drafted this decision to help you prepare a revised submission.

The reviewers were very enthusiastic about your work. They identified some shortcomings, but most of it could be addressed by text edits. The reviewers were less convinced about the envisioned link to reactive oxygen species (ROS). Ideally, you should consolidate this aspect by depicting the mis-regulated ROS in the mutant, and its restoration in the suppressor double mutants (e.g. by staining). However, the reviewers did not assume that these experiments are essential for acceptance. Alternatively, the discussion on this matter should be carefully revised.

Please, see the detailed comments of the reviewers below, which may guide you to further improve your manuscript.

Reviewer #1:

This study continues research started by professor Munetaka Sugiyama and his laboratory who identified about 20 years ago, or so, very interesting temperature-dependent fasciation (TDF) mutants affected in lateral root primordium (LRP) morphogenesis. The authors identified and report in this study genes responsible for the mutant phenotype of the root redifferentiation defective 1 (*rrd1*), *rrd2*, and *root initiation defective 4* (*rid4*). Intriguingly, all the genes are involved in RNA processing. Detailed analysis of the role of RRD2 and RID4 in mitochondrial mRNA editing and RRD1 in poly(A) degradation of mitochondrial mRNA make this work a solid and substantial study. The fact that pharmacological treatments of wild type seedlings by mitochondrial electron transport inhibitors can phenocopy the fasciated LRP phenotype is really fine. Similarly, the experiments with paraquat and ascorbate are very interesting. The main conclusion of the work (that LRP morphogenesis is linked to mitochondrial RNA processing and mitochondrion-mediated ROS generation) is novel and significant. I think this is an important step forward in our understanding the LRP morphogenesis.

I see only one main conceptual or interpretation problem.

The authors conclude that "that mitochondrial RNA processing is required for limiting cell division during early lateral root (LR) organogenesis". A similar statement appears where the authors postulate that TDF encode "negative regulators of proliferation that are important for the size restriction of the central zone during the formation of early stage LR primordia". Again, similar statements appear the Results, and in section of Discussion "Mitochondrial RNA processing is linked to the control of cell proliferation", especially where the authors say about "the control of cell proliferation at the early stage"

To my opinion, the above conclusions are arguable and cannot be accepted. To conclude about excessive cells division, the number of anticlinal divisions must be estimated per founder cell. This analysis has not been performed. The fact that at early stages LRPs are wider in the TDF mutants suggests that a greater number of FCs in the longitudinal plane participate in LRP formation. So, if this is correct, the mutations apparently affect control of lateral inhibition, and TDF genes are negative regulators of lateral inhibition. This question should be further investigated, but currently a more careful interpretation of the results is required. Also, if TDF genes encode "negative regulators of proliferation" then more frequent divisions would occur in the mutant. This question was not addressed either. If more frequent cell division is expected in early stage LRPs, this should result in formation of smaller cells. In accordance with Figure 1D of this study and Figures 1B and 3A of Otsuka and Sugiyama, 2012, this is not the case. Contrary, it seems that at the same developmental stage there are lower number of cells per unit of volume in the mutants compared to wild type. Another, possible explanation of the TDF mutant phenotype, in addition to lateral inhibition, is abnormal establishment of stem cell identity or affected stem cell function. Therefore, the mechanistic explanation of the link between TDF gene action and the respective mutant phenotype is not satisfactory. The interpretation given can be corrected and carefully rephrased throughout the text.

Reviewer #2:

The manuscript by Otsuka and co-workers, describes the mapping of the mutations in *rrd1*, *rrd2* and *rid4* causing the temperature sensitive lateral root morphogenesis defects (fascinated LR meristem). Interestingly, the respective mutated genes all map to genes involved in mitrochondrial mRNA processing, mRNA deadenylation, and mRNA editing. The authors propose that defective ROS homeostasis is causal to excessive cell proliferation in the lateral root primordia, and associated fasciation phenotype. Overall the manuscript is well-written, and is overall convincing with respect to characterization and mapping of the mutants, and the importance of RNA editing in mitochondria for the mutant phenotypes. I am not yet entirely convinced about the link to ROS production and the lateral root morphogenesis defects.

1) The fascinated LR phenotype is reminiscent of mutants defective in coordination of LR emergence, such as CASP:shy2 (Vermeer et al). Suggesting that defective signaling in LR overlaying layers, could be causal to the observed phenotype. However, the phenotyping presented in this manuscript does not allow to assess this. A detailed staging of LRPs would be required, and/or an analysis of the LRP developmental dynamics using a root bending assay.

2) Furthermore the expression domain analysis, shows clear expression in LRPs. However, I suspect expression of at least RID4-GFP in LRP overlaying layers. However, the resolution of the picture, and interference of the bright π counterstaining in Figure 2B preclude a thorough assessment of this.

3) The colocalization analysis in Figure 2D and E is not very clear. The mitotracker signal is set a bit too weak, making it difficult to assess the distinction between the GFP signal and the overlapping (yellow) signal). This could be amended by using different LUTs (also green/reds are not great for colorblind readers). Of note is the presence of relatively large structure labeled by RDD1-GFP, that is not colocalizing with mitotracker, suggesting it also localized to another subcellular compartment. Therefore, colocalization should be addressed more quantitatively, also using additional organellar markers. Additionally, the mitochondrial localization could be further supported by western blot on purified mitochondria.

4) The accumulation of polyadenylated transcripts in Figure 3D, seems to display also a temperature sensitivity in the WT. Why was this assay not done using a quantitative PCR, that will allow appreciating better the temperature component.

5) In contrast to the LR phenotyping as displayed in Figure 1, the LR phenotyping in Figure 4 is done in a completely different way. Why not use a uniform way to quantify. As it was done now, the suppression of rdd1 by ags1 mutation, is not very convincing, as the *rrd1* phenotype is nearly abolished in the Col-0 introgressed line (Figure 4 B), suggesting that the *rrd1* phenotype is sensitized in the Ler background.

6) While the authors focus on the LR morphology phenotype in the mutants, there is also a prominent effect on primary root growth that is not described. However, this phenotype does not seem to be very ecotype-specific, and is rescued in the ags1 background. A small phenotypic characterization of the primary root phenotype could thus be beneficial for the manuscript, and it wider relevance for development.

7) Figure 5. -> explain arrowheads in B, in the legend. Bar charts using mean + and – SD should be avoided when you do not have many data points, as in D and F (N=3 and 2). Better to show the raw data. Loading controls are missing for Figure 5 C and E.

8) The section about ROS is all based on ROS related pharmacology. However, ROS levels in the mutants were not assessed, making it difficult to use the pharmacological treatments to interpret the origin of the mutant phenotypes.

9) What is the link to the temperature sensitivity. Are these mutants hypersensitive to ROS inducing treatments?

10) While the role of ROS in LR development is key to the proposed model, the authors did not introduce what is the state of the art about ROS in lateral and primary root development.

11) In their model the authors might need to discuss whether or not ROS from the LRP could act as an intercellular coordinative developmental signal.

Reviewer #3:

Otsuka et al. report the characterisation of three temperature sensitive alleles of genes which prominently lead to overproliferation of cells in lateral root primordia. Interestingly this phenotype which is not underpinned by alteration of the auxin pattern, can be phenocopied by treatment with ROS and by interfering with the mitochondrial respiratory chain. This reveal that ROS modulate cell proliferation in the LR. The cloning and biochemical characterisation of the genes affected, reveal that all three encode enzyme involved in mt RNA processing, that perturb the production of certain components of the mitochondrial electron transport chain.

This is an excellent manuscript that points to a new and very interesting link between primary metabolism and cell proliferation in lateral roots. It is remarkably well written and presented. The conclusion are fully supported by the data. As it is the case for exciting new discoveries, they raise a lot of questions and this manuscript is no exception. It would be very interesting for future work to uncover the nature of the molecular link between ROS and cell proliferation and why are LR so sensitive to this. It'd be eventually interesting to speculate whether the reported existence of an hypoxic environment in the centre of the LRP has to do with this.

The one point, I would like to hear some comments from the authors about relates to the growth conditions used to reveal the phenotype at restrictive temperature. They mention that they use explant culture on RIM (characterised by high glucose and high 2.5µM IBA). What's the penetrance of the phenotype in standard (1/2 MS, 1% sucrose, no additional auxin/IBA)?

---

## [Author Response]

Reviewer #1:[…]I see only one main conceptual or interpretation problem.The authors conclude that "that mitochondrial RNA processing is required for limiting cell division during early lateral root (LR) organogenesis". A similar statement appears where the authors postulate that TDF encode "negative regulators of proliferation that are important for the size restriction of the central zone during the formation of early stage LR primordia". Again, similar statements appear in the Results, and in section of Discussion "Mitochondrial RNA processing is linked to the control of cell proliferation", especially where the authors say about "the control of cell proliferation at the early stage"To my opinion, the above conclusions are arguable and cannot be accepted. To conclude about excessive cells division, the number of anticlinal divisions must be estimated per founder cell. This analysis has not been performed. The fact that at early stages LRPs are wider in the TDF mutants suggests that a greater number of FCs in the longitudinal plane participate in LRP formation. So, if this is correct, the mutations apparently affect control of lateral inhibition, and TDF genes are negative regulators of lateral inhibition. This question should be further investigated, but currently a more careful interpretation of the results is required. Also, if TDF genes encode "negative regulators of proliferation" then more frequent divisions would occur in the mutant. This question was not addressed either. If more frequent cell division is expected in early stage LRPs, this should result in formation of smaller cells. In accordance with Figure 1D of this study and Figures 1B and 3A of Otsuka and Sugiyama, 2012, this is not the case. Contrary, it seems that at the same developmental stage there are lower number of cells per unit of volume in the mutants compared to wild type. Another, possible explanation of the TDF mutant phenotype, in addition to lateral inhibition, is abnormal establishment of stem cell identity or affected stem cell function. Therefore, the mechanistic explanation of the link between TDF gene action and the respective mutant phenotype is not satisfactory. The interpretation given can be corrected and carefully rephrased throughout the text.

Thank you for raising this important issue. We intended to use the phrase “excessive cell division” to refer to the increase in lateral root primordium cell number observed in the TDF mutants (Figure 1C to E), regardless of whether it is due to increased rounds of cell division of the founder cells (1) or participation of increased number of neighboring pericycle cells as founder cells to primordium formation (2). However, as you have pointed out, “excessive cell division” could be interpreted solely as (1). To avoid this confusion, we have replaced “excessive” with “extra” and also added a paragraph that states the two types of extra cell divisions ((1) and (2)) as a potential cause of LR fasciation to the Discussion section (sub-section “Extra cell division during early primordium development leads to LR fasciation”). With this correction, we believe that the statements made in our manuscript are now reasonable.

Reviewer #2:The manuscript by Otsuka and co-workers, describes the mapping of the mutations in rrd1, rrd2 and rid4 causing the temperature sensitive lateral root morphogenesis defects (fascinated LR meristem). Interestingly, the respective mutated genes all map to genes involved in mitrochondrial mRNA processing, mRNA deadenylation, and mRNA editing. The authors propose that defective ROS homeostasis is causal to excessive cell proliferation in the lateral root primordia, and associated fasciation phenotype. Overall the manuscript is well-written, and is overall convincing with respect to characterization and mapping of the mutants, and the importance of RNA editing in mitochondria for the mutant phenotypes. I am not yet entirely convinced about the link to ROS production and the lateral root morphogenesis defects.1) The fascinated LR phenotype is reminiscent of mutants defective in coordination of LR emergence, such as CASP:shy2 (Vermeer et al). Suggesting that defective signaling in LR overlaying layers, could be causal to the observed phenotype. However, the phenotyping presented in this manuscript does not allow to assess this. A detailed staging of LRPs would be required, and/or an analysis of the LRP developmental dynamics using a root bending assay.

Thank you for your valuable comment. As you have suggested, defective signaling in the LR overlaying tissues is one of the possible mechanisms that cause LR fasciation. However, given the current state of our research, it is beyond this work to determine the point of action of the TDF mutations. We would like to address this issue in future research by more detailed analysis of LRP development.

2) Furthermore the expression domain analysis, shows clear expression in LRPs. However, I suspect expression of at least RID4-GFP in LRP overlaying layers. However, the resolution of the picture, and interference of the bright π counterstaining in Figure 2B preclude a thorough assessment of this.

Thank you for comment. We had mentioned weak expression of the GFP markers in tissues overlaying the LR primordium in the original manuscript; however, as you have pointed out, the quality of the original micrograph was not satisfactory to make a clear assessment. We have redone the microscopy with some technical improvements, such as the use of a 60X oil emulsion objective lens instead of a 40X water emulsion objective lens and weaker π counterstaining. The new micrographs (Figure 2B) provide better resolution of the GFP expression patterns.

3) The colocalization analysis in Figure 2D and E is not very clear. The mitotracker signal is set a bit too weak, making it difficult to assess the distinction between the GFP signal and the overlapping (yellow) signal). This could be amended by using different LUTs (also green/reds are not great for colorblind readers). Of note is the presence of relatively large structure labeled by RDD1-GFP, that is not colocalizing with mitotracker, suggesting it also localized to another subcellular compartment. Therefore, colocalization should be addressed more quantitatively, also using additional organellar markers. Additionally, the mitochondrial localization could be further supported by western blot on purified mitochondria.

Thank you for your comments and helpful suggestions. We have adjusted the mitotracker signal level and also converted the red channel to magenta. We have also added data from experiments using protoplasts derived from greening-induced callus to check for plastid localization of RRD1:GFP and RID4:GFP (Figure 2—figure supplement 5).

4) The accumulation of polyadenylated transcripts in Figure 3D, seems to display also a temperature sensitivity in the WT. Why was this assay not done using a quantitative PCR, that will allow appreciating better the temperature component.

Thank you for your comment. Although qPCR is effective in quantifying poly(A)+ mRNA, the RACE-PAT assay used in Figure 3D and Figure 4A is also considered an established method in the relevant field and well suited for the purpose of our study.

5) In contrast to the LR phenotyping as displayed in Figure 1, the LR phenotyping in Figure 4 is done in a completely different way. Why not use a uniform way to quantify. As it was done now, the suppression of rdd1 by ags1 mutation, is not very convincing, as the rrd1 phenotype is nearly abolished in the Col-0 introgressed line (Figure 4B), suggesting that the rrd1 phenotype is sensitized in the Ler background.

Thank you for your comment. Although the *rrd1* LR phenotype is weaker in the Col background, we observed a statistically significant decrease of the LR width in the *rrd1 ags1* double mutant, compared to the control (*rrd1 AGS1^C^*). Throughout this study, we have used the measurement of LR width at the 6^th^ day of culture as the standard method to assess the LR phenotype. In the cytological characterization of LR fasciation in the TDF mutants shown in Figure 1, we employed a different observation protocol in order to measure the cell number of the outermost layer of MOL and the widths of MOL and MTL at particular stages.

6) While the authors focus on the LR morphology phenotype in the mutants, there is also a prominent effect on primary root growth that is not described. However, this phenotype does not seem to be very ecotype-specific, and is rescued in the ags1 background. A small phenotypic characterization of the primary root phenotype could thus be beneficial for the manuscript, and it wider relevance for development.

Thank you for your constructive comment. In the original manuscript, the phenotype shown in Figure 4C was only mentioned as “seedling growth retardation”. We have slightly modified the text to explicitly refer to the root and shoot growth phenotype and additionally cited our previous report on the seedling phenotypes of the *rrd1* mutant.

7) Figure 5. -> explain arrowheads in B, in the legend. Bar charts using mean + and – SD should be avoided when you do not have many data points, as in D and F (N=3 and 2). Better to show the raw data. Loading controls are missing for Figure 5C and E.

Thank you for your comments. We have added explanations regarding the arrowheads in Figure 5B in the legend and also added explanation on mitochondrial complex identification in the Materials and methods section. We have also added the raw data points to the bar graphs of Figure 5D and F. For the western blots, we ensured accurate adjustment of the loaded samples by employing a modified version of the Bradford method (XL-Bradford), which allows direct quantification of the SDS-PAGE samples. In each experiment, the accuracy of the protein quantification by XL-Bradford was additionally checked by observation of CBB stained electrophoresis gels (as shown in Author response image 1 for Figure 5C); however, for Figure 5E, we unfortunately failed to obtain a digital image data of the stained gel, due to physical damage to the gel.

8) The section about ROS is all based on ROS related pharmacology. However, ROS levels in the mutants were not assessed, making it difficult to use the pharmacological treatments to interpret the origin of the mutant phenotypes.

Thank you for your comment. We agree with you opinion that it is highly important to quantify ROS levels in the TDF mutants; however, we found it very difficult to accurately assess mitochondrial ROS in the LR primordium, due to issues such as the impermeability of the root endodermis to the fluorescent probes and the lack of adequate techniques to distinguish mitochondrial ROS from apoplastic ROS when using colorimetric methods. Further efforts would be required to improve the situation.

9) What is the link to the temperature sensitivity. Are these mutants hypersensitive to ROS inducing treatments?

Thank you for your questions. To reinforce the Discussion on this issue in the sub-section “The origins of temperature sensitivity may differ among the TDF mutants”, we have added data concerning the combined effects of mitochondrial respiratory inhibition and high temperature on LR fasciation (Figure 6—figure supplement 1) and also some arguments.

10) While the role of ROS in LR development is key to the proposed model, the authors did not introduce what is the state of the art about ROS in lateral and primary root development.11) In their model the authors might need to discuss whether or not ROS from the LRP could act as an intercellular coordinative developmental signal.

Thank you for your important comments. We have added a paragraph dedicated to the issues raised in 10) and 11) in the sub-section “Impaired mitochondrial electron transport causes LR fasciation likely via ROS production” of the Discussion section.

Reviewer #3:[…]The one point, I would like to hear some comments from the authors about relates to the growth conditions used to reveal the phenotype at restrictive temperature. They mention that they use explant culture on RIM (characterised by high glucose and high 2.5µM IBA). What's the penetrance of the phenotype in standard (1/2 MS, 1% sucrose, no additional auxin/IBA)?

Thank you for your question. In order to examine the effects of the TDF mutations on mophogenesis of new LR primordia, we used the semi-synchronous induction system throughout this study. When seedlings of the TDF mutants were grown in the standard condition at the restrictive temperature, it was difficult to see the direct effects of the mutations on LR development because primary root growth was severely affected as shown in our previous papers and also in Figure 4C and Figure 2—figure supplement 3B of this work. When seedling grown in the standard condition at the permissive temperature were exposed to the restrictive temperature by a simple temperature-shift protocol, we occasionally observed formation of abnormal LRs including fasciated ones. However, we did not evaluate quantitatively this phenotype because we could not determine which LRs were newly formed and which LRs were developed from the existing primordia. Examination of LRs induced via gravistimulated root bending in a temperature-shift experiment could be suitable for answering your question, since the LR primordia that have experienced the temperature-shift are expected to be easily identifiable. Preliminary experiments using *rid4-2* have suggested that fasciation also occurs in root bending-induced LRs, but further refining of the experimental system is required.